# A sensory memory processing system with multi-wavelength synaptic-polychromatic light emission for multi-modal information recognition

Liuting Shan[1,2], Qizhen Chen[1,2,3], Rengjian Yu[1,2], Changsong Gao[1,2], Lujian Liu[1,2], Tailiang Guo[1,2] & Huipeng Chen [1,2] ✉

Realizing multi-modal information recognition tasks which can process external information efficiently and comprehensively is an urgent requirement in the field of artificial intelligence. However, it remains a challenge to achieve simple structure and high-performance multi-modal recognition demonstrations owing to the complex execution module and separation of memory processing based on the traditional complementary metal oxide semiconductor (CMOS) architecture. Here, we propose an efficient sensory memory processing system (SMPS), which can process sensory information and generate synapse-like and multi-wavelength light-emitting output, realizing diversified utilization of light in information processing and multi-modal information recognition. The SMPS exhibits strong robustness in information encoding/transmission and the capability of visible information display through the multi-level color responses, which can implement the multi-level pain warning process of organisms intuitively. Furthermore, different from the conventional multi-modal information processing system that requires independent and complex circuit modules, the proposed SMPS with unique optical multi-information parallel output can realize efficient multi-modal information recognition of dynamic step frequency and spatial positioning simultaneously with the accuracy of 99.5% and 98.2%, respectively. Therefore, the SMPS proposed in this work with simple component, flexible operation, strong robustness, and highly efficiency is promising for future sensory-neuromorphic photonic systems and interactive artificial intelligence.

A large number of computational operations, such as multiply and accumulate (MAC) in the traditional way, have become a serious burden on current central processing units (CPUs)[1]. Inspired by the working mode of the human brain, the idea of enabling neural computing at the synaptic level has attracted great interest[2–12]. Currently, neuromorphic synapse devices and neuromorphic systems have been widely used in bioelectronics and artificial intelligence computing[13–17]. Future neuromorphic engineering holds promise for

[1]Institute of Optoelectronic Display, National & Local United Engineering Lab of Flat Panel Display Technology, Fuzhou University, Fuzhou 350002, China. [2]Fujian Science & Technology Innovation Laboratory for Optoelectronic Information of China, Fuzhou 350100, China. [3]School of Opto-electronic and Communication Engineering, Xiamen University of Technology, Xiamen 361024, China. ✉e-mail: hpchen@fzu.edu.cn

intelligent and humanized systems that interact directly with humans. However, most of the current work reported on neuromorphic devices or systems is limited to a single pattern recognition task such as MNIST handwritten digit recognition, etc., limiting future advances to a wide range of applications such as prosthetics, robotics, and cyborg systems[18–21].

With the development of the Internet of Things era, the requirements of the application scenarios and modal recognition of neuromorphic devices/systems tend to be diversified, and multi-modal/multi-task recognition has become an urgent task to improve the current new hardware architecture of artificial intelligence[22–26]. Multi-modal sensing and learning usually obtain data from different modes by elaborating or describing the same entity or spatiotemporal event in different modal spaces. These multi-modal descriptions can depict the multi-dimensional information of the same objective entity more comprehensively so as to improve the understanding and cognitive ability of the model. In the reported work, multimodal devices based on the detection of multimodal signals have been proposed for the development of artificial intelligence (AI) for health monitoring and environmental interaction[27–33]. However, these electronic devices often require multiple input terminals for the acquisition of multiple modal information. Moreover, coupled auto-en/decoder and necessary peripheral circuit modules are required to realize the feature extraction and multimodal data fusion process, which imposes a significant burden on the production costs of large-scale applications[30,34,35]. Recently, multimodal sensors capable of measuring more than two stimuli in a single unit have attracted researchers' interest[29,36–38]. However, such multimodal sensors containing different functional layers often inevitably produce reciprocal interfering electrical signals, which will have indistinguishable consequences for subsequent preprocessing processes in neuromorphic devices.

Optical information has opened a new door for information security communication and storage due to its rich degree of freedom characteristics[39–41]. Existing studies have shown that parameters such as wavelength, amplitude, phase, and polarization state of light can be used as carriers to realize information loading[42,43]. Therefore, it will be a promising solution to apply the multi-information loading characteristic of light to the multi-modal recognition field. In addition, with the maturity of wavelength division multiplexing (WDM) and high-speed modulation technology, optical methods can parallelize data transmission and processing at the speed of light[44–48]. Using an all-optical pulse scheme, the optical transmission through PCM-based memristors can be gradually tuned, enabling high-speed and large bandwidth photonic in-memory computation[49]. By combining the optical implementation method with the artificial neural networks (ANN) architecture, is expected to provide a better computing platform for AI. The optical stimulation of memristors is used as an independent signal to trigger a more linear and symmetric switching behavior, which could allow high-density, energy-efficient neuromorphic computing chips[10]. Furthermore, the implementation of a "photon brain" with powerful sensing and processing capabilities that can learn from the output and generate experience is expected to build an advanced architecture for future neuromorphic processing systems (Fig. 1a).

Here, a strategy is proposed to exploit a sensory memory processing system with synaptic-polychromatic light emission utilizing a triboelectric nanogenerator (TENG) as a sensory receptor and quantum-dot light emitting diode (QLED) device as a light-emitting artificial neuromorphic synapse, which can achieve synapse-like and multi-wavelength optical signal output and implement multimodal information recognition through an artificial neural network (Fig. 1b). The optical output signal enables robust information encoding and transmission, intelligent decision processing, and human–machine interaction. By collecting the wavelength-amplitude parameters of the optical signal at the same time, the spatial positioning and dynamic step frequency multi-modal recognition are successfully realized with high accuracy. Different from the conventional multi-modal information processing system, which requires a different mode sensing module, independent memory processing module, and complex coupled auto-en/decoder module, the proposed sensory memory processing system with multi-information parallel output can realize the function of multi-module integration, which greatly simplifies the complexity of the circuit (Fig. 1c). These achievements are expected to pave the way for a new generation of intelligent optical communication and neuromorphic photonic system.

## Results

### The fundamental properties of TENG and QLED

Single-electrode mode TENG (S-TENG) acts as a receptor to collect sensory signals contact-separated from the skin and convert them into presynaptic pulses. The voltage signals are then transmitted to an artificial synaptic device implemented by QLED to generate electroluminescence and postsynaptic current (PSC). Electroluminescence is received by the photodetector for postsynaptic brightness (PSB) detection. Hybrid quantum dots (QDs) used in the light-emitting layer can emit light of specific color in response to electric field[7,50–53], and the emission color can be changed by varying the magnitude of the electric field. Based on this, SMPS with synapse-like and multi-wavelength light output can be realized by tuning the electric field intensity generated by the contact intensity. Figure 2a shows the schematic structural diagram of the TENG. The single-electrode mode of the TENG induces the triboelectric potential from the contact of the friction layer PDMS with the skin. The organic polymer film PDMS is used as the friction layer, the Ag NWs as the flexible electrode, and the PDMS as the protective layer substrate to prepare a flexible sensory sensor. The effects of the two layers of PDMS are different. The PDMS of the upper layer is used as the friction layer due to its good electronegativity, which can generate charge transfer when separated in contact with the finger. Since PDMS has good stretchability and can be used as a flexible substrate to attach to the surface of the untitled/human body, the PDMS in the lower layer is used as the substrate of the whole TENG to provide wearable function. A continuous voltage response is achieved by contact separation from the skin, which is a process according to contact-electrification and electrostatic induction, and the specific operating principle is shown in Supplementary Fig. S1. The open-circuit voltage of the TENG at different pressures and different frequencies is measured (Fig. 2b, c). When the pressure increases from 4 to 7 kPa, the corresponding open-circuit voltage increases from about 13 to 30 V (Fig. 2b). Figure 2c plots the open-circuit voltage when the frequency varies from 1 to 4 Hz at the pressure of 6 kPa. One working cycle of TENG is shown in Supplementary Fig. S2, in which the rise time and fall time of the releasing signal are 3 and 1 ms, and the rise and fall of the contacting signal are 1 and 3 ms, respectively. The corresponding voltage signal increases as the pressure increases, and the voltage saturates at 40 V when the pressure approaches 20 kPa (Supplementary Fig. S3). This result can be attributed to the fact that the increased pressure can achieve more contact area and thus increase the surface charge density, enabling higher voltage output. Moreover, the performance of TENG is compared to TENG with the same structure and different electrode materials or different friction layer materials (Supplementary Fig. S4). The PDMS/AgNWs/PDMS single-electrode TENG exhibits good performance because of the excellent electrical conductivity of silver nanowires and the good electronegativity of PDMS. The structure of the QLED device consists of ITO/PEDOT:PSS/Poly-TPD hole transport layer (HTL)/PVP trapping layer/red-green hybrid QDs emission layer (EML)/ZnO NPs electron transport layer (ETL)/Ag. The energy level diagram of the device is shown in Fig. 2e. The QDs emission layer is fabricated by mixing red and green quantum dots in a mass ratio of 15:10. Figure 2f shows an atomic force microscope (AFM) image of the QDs layer film with a root-mean-square

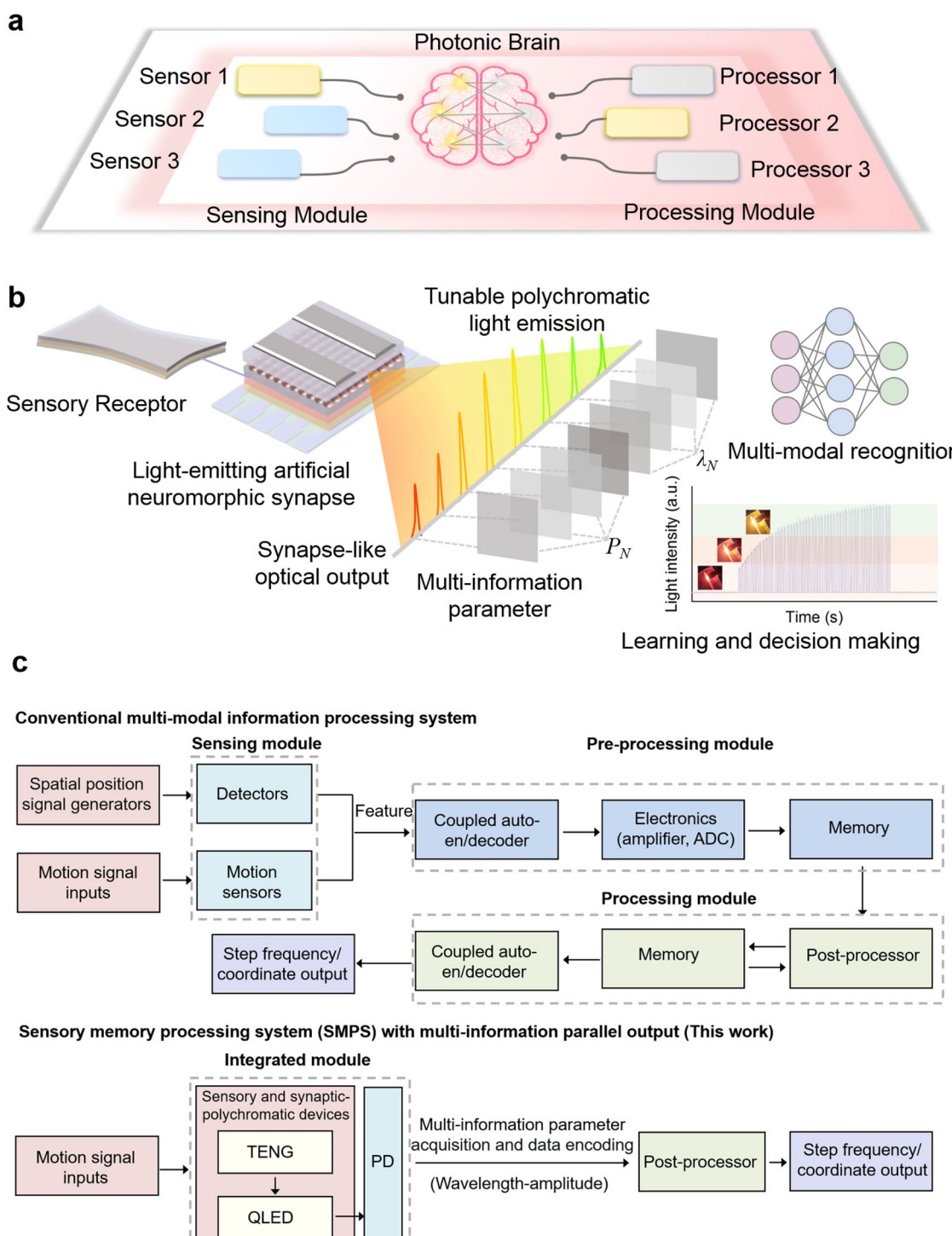

**Fig. 1 | Schematic diagram of the photonic brain and the SMPS. a** A "photonic brain" with sensing modules and processing modules is the ultimate goal of future neuromorphic photonic systems. **b** Sensory memory processing system with synapse-like optical output, polychromatic light emission, and multi-information parameter loading (wavelength amplitude), which can realize learning/decision making and multi-modal recognition. **c** Block diagrams of the conventional multi-modal information processing system and SMPS with multi-information parallel output.

(RMS) roughness of 3.24 nm. Meanwhile, the AFM image of PVP film and ZnO NPs films with the RMS roughness of 0.408 and 5.56 nm are shown in Supplementary Fig. S5. Figure 2d illustrates the current–voltage curve of the device from 0 to 6 V scan voltage, showing typical diode characteristics. The inset in Fig. 2d depicts that the transient luminescence brightness of the QLED device gradually increases with a significant color change when the applied voltage exceeds the turn-on voltage of the device. Correspondingly, the well-behaved voltage–luminance curve of the device under forward bias proves that the device has good luminescence behavior, and the external quantum efficiency (EQE), current efficiency (CE), and power efficiency (PE) of the device are also investigated as a function of current density (Supplementary Fig. S6). The electroluminescence (EL) spectrum of the device at different biases is further measured (Fig. 2g).

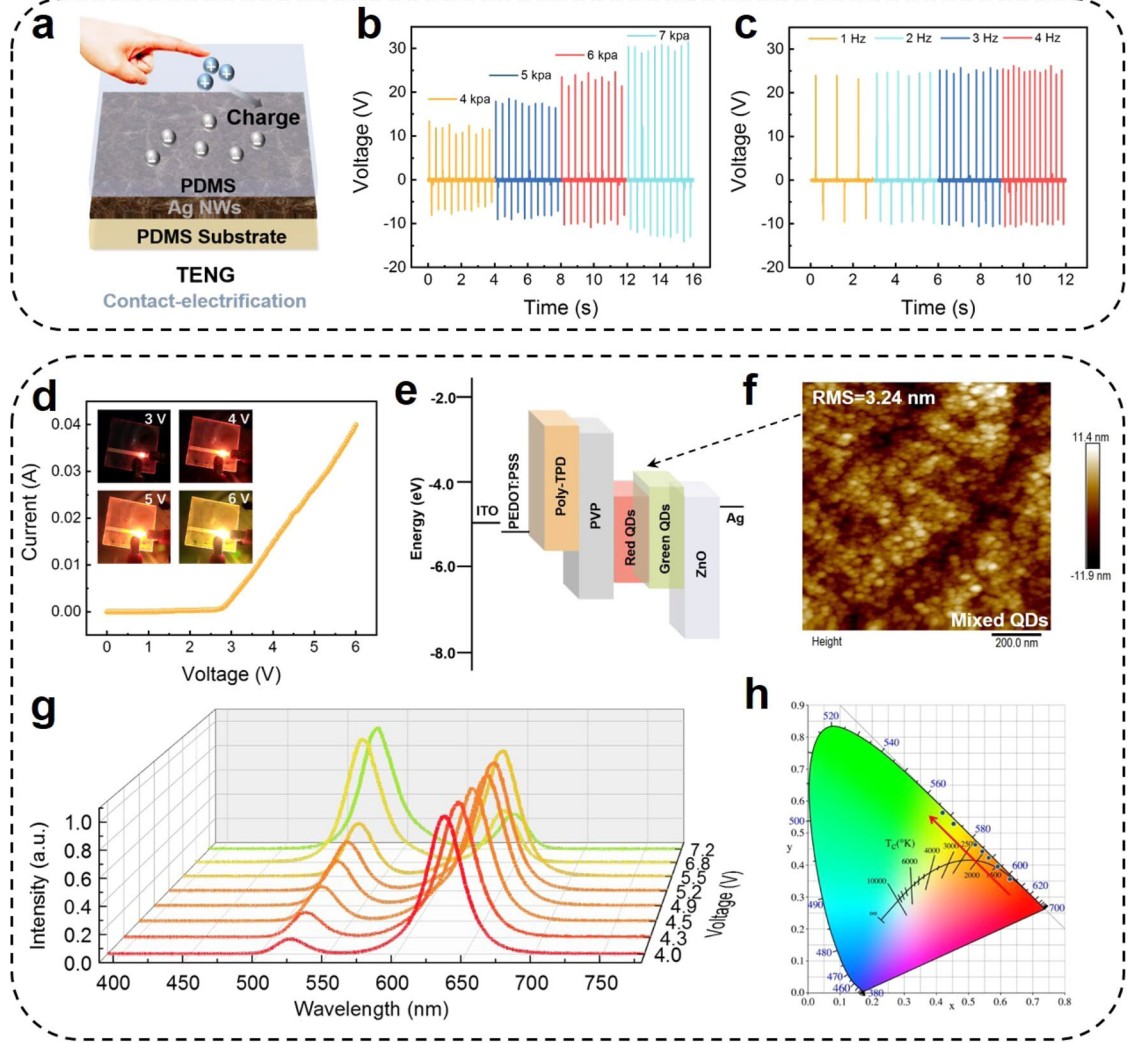

**Fig. 2 | The fundamental properties of TENG and QLED. a** Schematic diagram of the structure of TENG. The blue balls represent the holes, and the gray ones represent the electrons. **b** Open-circuit voltage ($V_{OC}$) of TENG under various pressures from 4 to 7 kPa and **c** frequencies from 1 to 4 Hz. **d** $I–V$ characteristics of QLED measured in scan voltage from 0 to 6 V (the inset is an optical photograph of working QLED under different voltage). **e** The energy level diagram of the QLED device. **f** The atomic force microscope (AFM) image of the hybrid QDs layer film. **g** The EL optical spectra of the QLED under different voltages. **h** The CIE 1931 coordinates diagram of the multicolor QLED.

When the voltage increases from 4.0 to 7.2 V, the dominant color changes from red (630 nm) to green (525 nm), following the preferential occupation of red quantum dots with a higher valence band and narrower bandgap by electrons and holes[51]. As the injection current increases, carriers are injected into the green quantum dots, showing a color change from red to green. This trend of color change is also reflected in the chromaticity coordinates (CIE 1931) diagram (Fig. 2h).

### Characterization of synaptic properties of QLED

Figure 3a depicts a schematic diagram of the artificial light-emitting synapse device, which simulates a biological synapse in a sensory system. Typical excitatory postsynaptic current (EPSC) properties and corresponding excitatory postsynaptic brightness (EPSB) are exhibited in our device. The common electrical output behaviors of synaptic devices under the stimulation of circulating electrical pulses are demonstrated (Supplementary Figs. S7 and S8) due to the trapping of injected holes in the PVP layer under an external electric field, which will be discussed in detail below. Figure 3b shows the $I–V$ characteristics of the QLED during thirty consecutive positive and negative voltage sweeps ($-6 \rightarrow 0 \rightarrow 6 \rightarrow 0 \rightarrow -6$ V). As the number of voltage sweeps increases, the conductance of the synaptic device increases continuously and tends to saturation, and finally, a stable $I–V$ curve is obtained. Similarly, we investigate the electrical output synaptic properties of the device at different voltage amplitudes. When a positive voltage pulse with an amplitude of 3 V is applied, the device current is essentially unchanged. However, the maximum current of the device increases by a factor of 19.1 as the amplitude of the applied electrical stimulation continues to increase (Fig. 3c). The light output time response of the device is illustrated when a square wave pulse bias at $V_{in}$ (6 V amplitude, 90 ms pulse width) is used, where the rise and fall times are $\tau_{rise}$ ~25 ms and $\tau_{fall}$ ~11 ms (Supplementary Fig. S9). This demonstrates the instantaneous response of QLED's light emission to pulse bias, and parallel data transmission using the light-emitting synapse device is indeed feasible. Paired-pulse facilitation (PPF) is an important feature of short-term synaptic plasticity. The PPF index of output current and light signal as a function of $\Delta t$ is plotted, and the results are consistent with the PPF behavior of synapse (Supplementary Fig. S10).

The responses of EPSC and EPSB to pulse stimulation with different frequencies and numbers are further investigated. The frequency responses of EPSC and EPSB of the device increase steadily

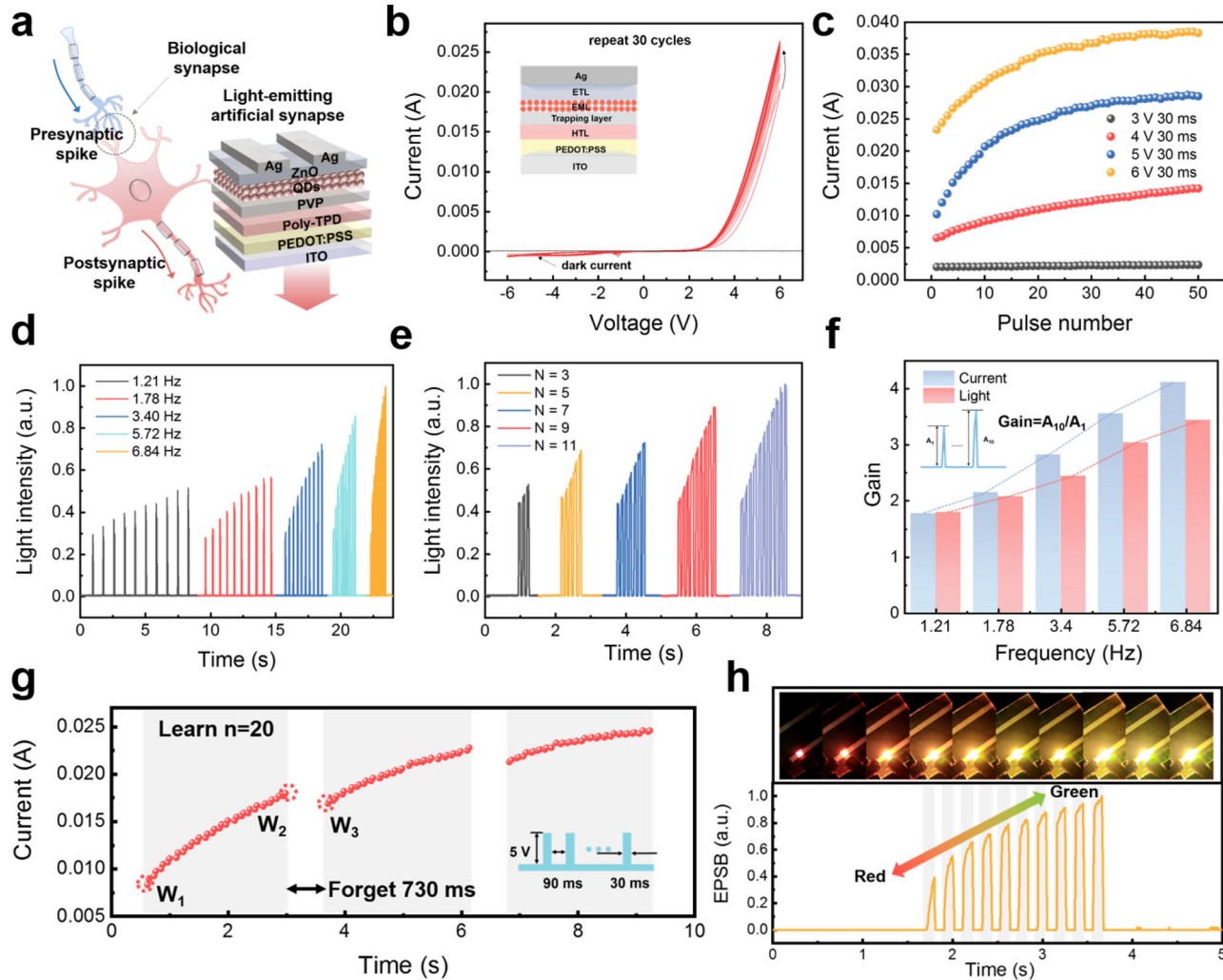

**Fig. 3 | Characterization of synaptic properties of QLED. a** Schematic diagram of the artificial light-emitting synapse device simulated by QLED. **b** I–V characteristics of the QLED during thirty consecutive voltage double sweeps (−6 → 0 → 6 → 0 → −6V). **c** The electrical output synaptic properties of the device at different voltage amplitudes (50 pulse number, 30 ms duration). **d** The response of EPSB to pulse stimulation with different frequencies from 1.21 to 6.84 Hz and **e** numbers from 3 to 11. **f** EPSB and EPSC gain ($A_{10}/A_1$) (determined by the ratio of the 10th EPSC peak ($A_{10}$) to the first EPSC peak ($A_1$)) of the artificial synapse from 1.21 to 6.84 Hz (6 V amplitude, 30 ms duration). **g** The electrical synaptic property simulates the learning and forgetting process in the human brain. **h** The light output characteristic and luminance color change of QLED synapse under continuous stimulation.

from 1.21 to 6.84 Hz (6 V, 30 ms) (Fig. 3d and Supplementary Fig. S11). The EPSC gain (determined by the ratio of the 10th EPSC peak $A_{10}$ to the first EPSC peak $A_1$) gradually increases from 1.77 to 4.12, while the EPSB gain gradually increases from 1.79 to 3.43 (Fig. 3f). Figure 3e and Supplementary Fig. S12 depict the output response of the device when the number of pulses is increased from 3 to 11 pulses (6 V, 30 ms). The results show that EPSC and EPSB exhibit the same increasing trend due to the accumulation of carriers. Similarly, EPSC and EPSB gains vary almost proportional to frequency (from 1.28 to 2.97 and 1.19 to 2.16, respectively) (Supplementary Fig. S13). The amplitude and pulse width response of the device is also investigated, as seen in Supplementary Figs. S14 and S15.

Figure 3g shows the memory properties of QLEDs, which are represented by QLED typical learning and forgetting features. The initial value is recorded as $W_1$. When a continuous pulse (5 V, 30 ms, N = 20) is applied, the postsynaptic current continues to increase to a value of $W_2$. At this time, the pulse is removed for a period of time (forgetting 730 ms), and the current value $W_3$ is measured below $W_2$; the process is repeated thereafter, and the output current continues to increase until saturation. This result demonstrates that our device can simulate the human brain's processing of events, including repetitive

learning, memory, and forgetting. Figure 3h illustrates the light output characteristic and luminance change of QLED synapse under continuous stimulation. As mentioned earlier, the light-emitting synapse exhibits a color change from red to yellow to green, which is due to the tendency of low energy emission that appears first, followed by high energy emission under successive pulses. This result proves that it is feasible for our device to realize optical signal transmission/communication with autonomous learning capability.

## The operation mechanism of SMPS

Figure 4a–c illustrates the working mechanism of the SMPS. TENG is acted as a receptor to collect human motion signals, and QLED is used as a synapse to analyze and process the collected signals. Here, the TENG is integrated with the QLED, and the electrode of the TENG is connected to the anode ITO of the synaptic device, delivering the output voltage signal as a presynaptic potential. On the left of Fig. 4a, the skin and the PDMS film are in contact with each other. Electrons are conducted from the skin to the PDMS surface due to electrostatic induction. Under the effect of electrostatic equilibrium[54], electrons are compensated from the ITO electrode to the Ag NWs electrode, resulting in the presynaptic peak of the QLED device. Under positive

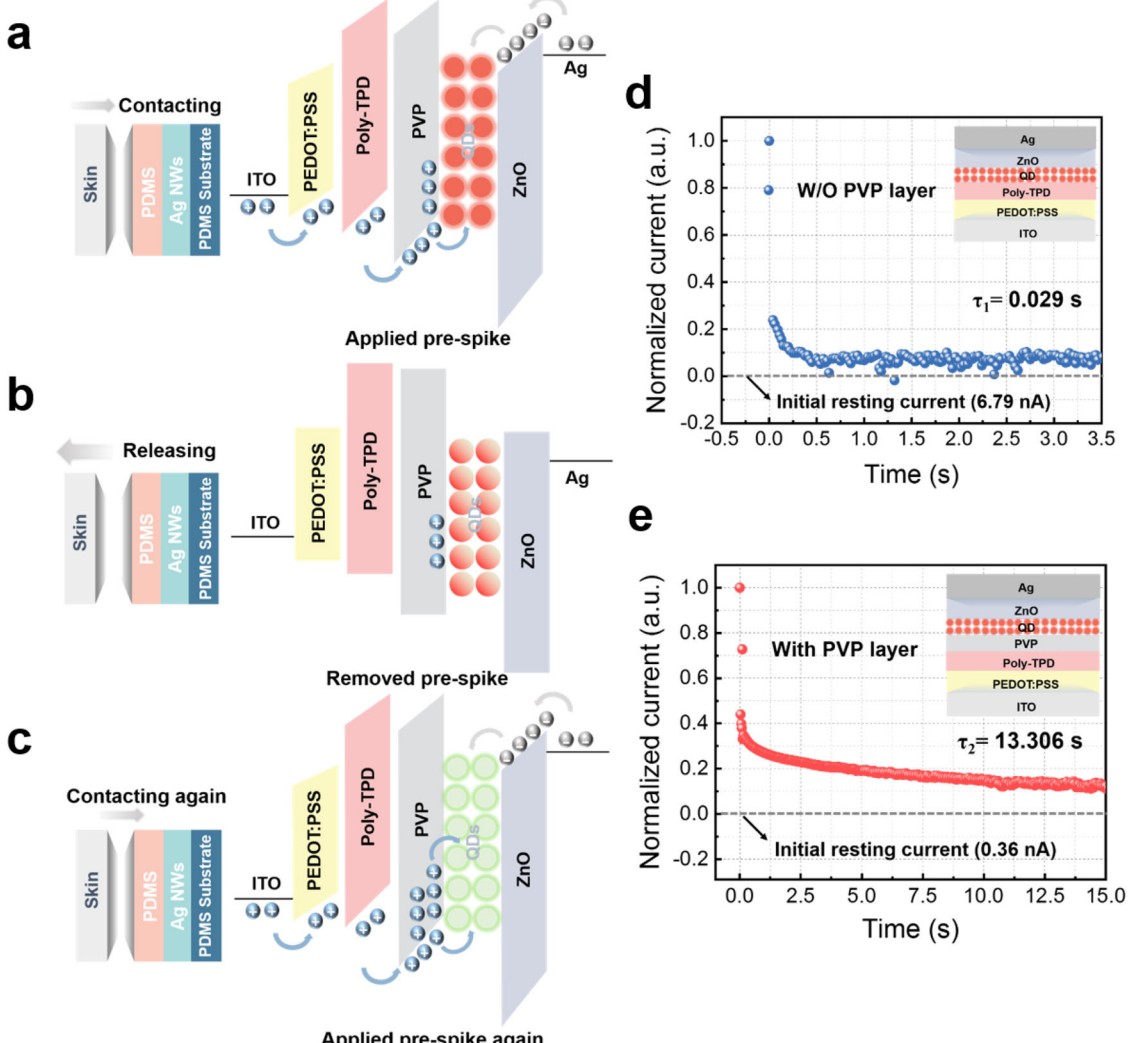

**Fig. 4 | The operation mechanism of SMPS. a–c** Working mechanism of the sensory memory processing system. Schematic illustrations of the working principles at **a** contacting, **b** releasing, and **c** contacting again. The blue balls represent the holes, and the gray ones represent the electrons. **d** Long-term decay time curve of the output current of the synaptic device without the PVP layer (initial resting current at 6.79 nA) and **e** with the PVP layer (initial resting current at 0.36 nA).

spike stimulation, electrons and holes are injected from the electrodes at both ends and transferred to the QD light-emitting layer through the charge transport layers for radiative recombination to generate photons. The properties of the artificial synapse are achieved by embedding the trapping layer PVP into QLED. In general, PVP is a dielectric material containing deep trap-derived, abundant ionic groups that allow for the trapping and de-trapping of charge carriers upon application of a voltage[55,56]. The working mechanism of the PVP charge-trapping layer is shown in Supplementary Fig. S16. Therefore, a part of the holes can be captured by the PVP layer during the first voltage stimulation. Then, when the pressure is released, as shown in Fig. 4b, the external voltage of the QLED is restored to 0 V, resulting in an instantaneous decrease in EPSC while the holes are stored in the PVP layer. When the two surfaces are in contact again, the trapped holes are released under the action of an applied electric field, thereby increasing the electrical conductivity and EPSB of the device, making the color tunability more obvious (Fig. 4c). Applying continuous write-erase pulses to the control group with and without the PVP layer, the conductance change with the PVP layer is bigger than that of the device without the PVP layer, the comparison in conductance change also proves that the PVP trapping mechanism is reasonable (Supplementary Fig. S17). To demonstrate the memory properties of the synaptic device, we compare the long-term decay time curves of the

devices with the PVP layer and the control (the ordinate is normalized by the log of the current) (Fig. 4d, e and Supplementary Fig. S18). According to the exponential decay function:

$$Y = Ae^{(-x/\tau)} + y \tag{1}$$

where $A$ is the prefactor and the relaxation times $\tau_1 = 0.029$ s and $\tau_2 = 13.306$ s of the control, and the device with the PVP layer are fitted, respectively. The difference in relaxation time indicates that the presence of the PVP trapping layer can retain the charge after removing the bias, and the conductance value will not decay to the initial state in a short time. This clearly illustrates that our device has a similar tendency to a long, slow decline in human memory following an initial rapid decline in a biological system. These results indicate that QLED devices with a PVP charge trapping layer have better synaptic performance and are more suitable to be integrated into our SMPS as light-emitting synaptic devices.

## Multiparameter information optical communication and visible multi-level injury warning

To demonstrate the advantages of our system, an intelligent optical communication application based on wireless transmission and reception is designed (Fig. 5a). The QLED synapse based on the

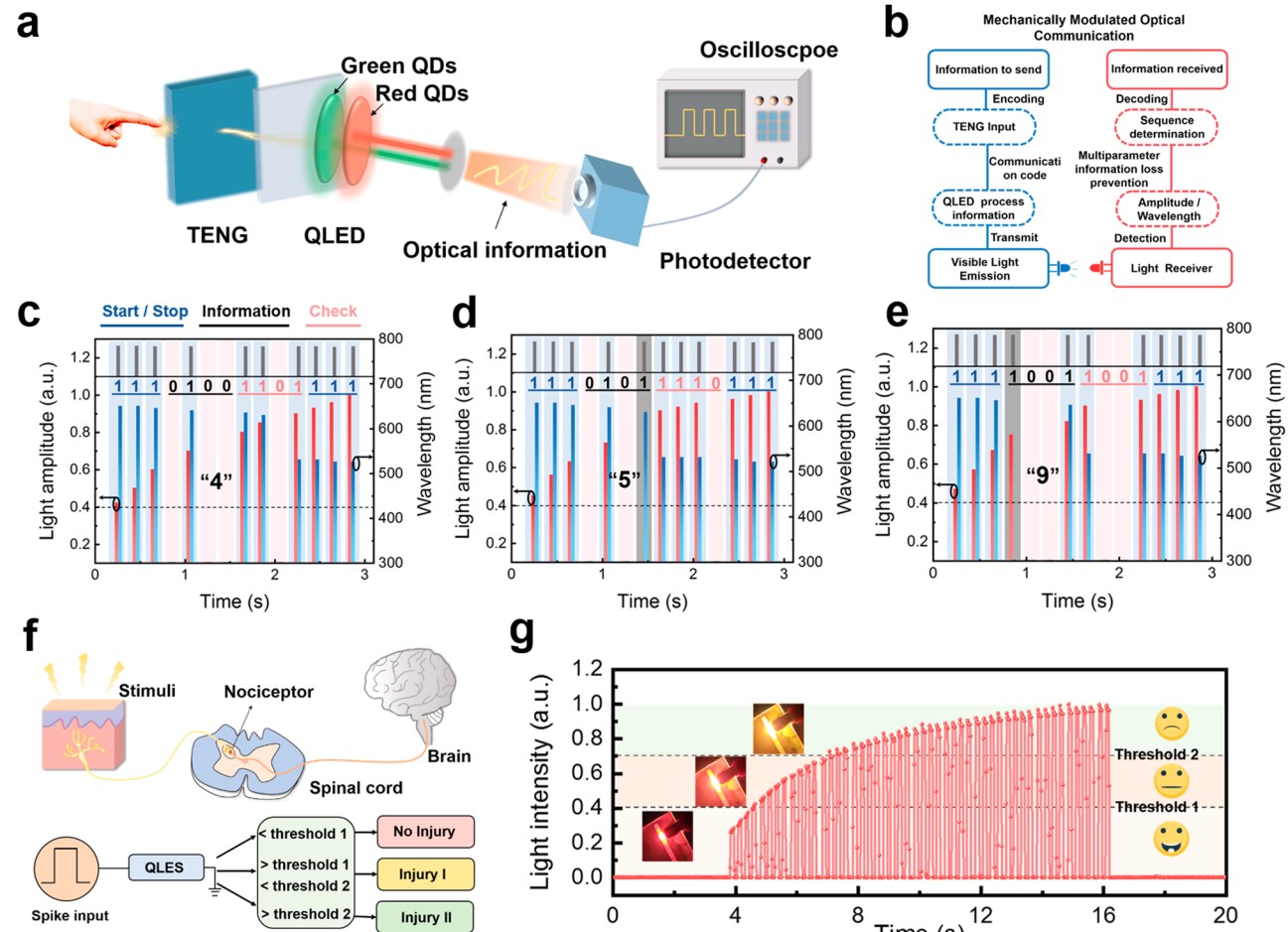

**Fig. 5 | Multiparameter information optical communication and visible multi-level injury warning. a** Schematic diagram of intelligent optical communication based on wireless transmission and reception. **b** The flow chart for mechanically modulated optical communication. **c–e** The corresponding light amplitude intensity and wavelength (dominant wavelength) readings when encoding/decoding numbers 4, 5, and 9 in optical communications (the shaded parts in the figure represent the loss of **d** amplitude information and the loss of **e** wavelength information). **f** Schematic diagram of SMPS to realize biological pain perception and make a multi-level warning process. **g** Under continuous stimulation, the system realizes multi-level injury warning response and visible luminous color change.

mixture of red-green quantum dots is regulated by TENG to emit light, and the emitted optical signal is detected by a photodetector and converted into an electrical signal to be transmitted to the oscilloscope for analysis. Figure 5b shows a flow chart for mechanically modulated optical communication. At the sending end, the message to be sent is first encoded into a sequence as the input of the TENG, which represents the communication code transmitted to the QLED for visible light emission. At the receiving end, information such as the amplitude/wavelength of the visible light signal is first detected by the receiver. Then, the light multiparameter information is linearly mapped into sequences. Finally, the sequence is decoded to read the information. Figure 5c–e shows the corresponding optical amplitude intensity and wavelength (dominant wavelength) reading when the three codes achieve optical communication. Here, the bit sequence consists of three parts: start/stop, information, and check bits[38,57]. Taking Fig. 5c as an example, the peak in the output that exceeds the given threshold corresponds to '1'; otherwise, it corresponds to '0'. Here, we set the threshold value at 0.4, corresponding to the normalized value of the current intensity and the threshold value at 425 nm corresponding to the wavelength. With this concept, the communication sequence can be decoded. According to the multi-parameter characteristics of optical information, optical amplitude, and wavelength can be output in parallel, avoiding decoding errors caused by the loss of information. The gray shaded

parts represent the normal information transmission when only wavelength information (Fig. 5d) and amplitude information (Fig. 5e) exist, which reflects the stronger robustness of optical information as a communication medium. As the next generation of intelligent optical communication systems, it should have the function of sensing external stimuli and making intelligent decision processing similar to the biological system. Figure 5f shows the schematic diagram of our system to realize biological pain perception and a multi-tilevel warning process. In biological systems, nociceptors receive signals of noxious stimuli and compare the amplitude of the signal with its threshold to determine whether an action potential is generated and sent to the brain via the spinal cord (central nervous system)[58]. In the receptor simulated by TENG, if the input external stimuli are not strong enough or the number is small, the QLED will present low-energy red light emission and low-intensity output, indicating that the external stimuli are harmless (No injury). Here, since three levels of damage can be judged, two thresholds are divided at 0.4 and 0.7 of the normalized current intensity. When the input external stimulus continuously increases, the induced light pulse amplitude will be larger than the threshold, which is attributed to the continuous accumulation of carriers to generate more recombination luminescence. The device color is observed at the output as orange (Injury I) or green (Injury II), a process that corresponds to pain perception of noxious stimuli. The light emission

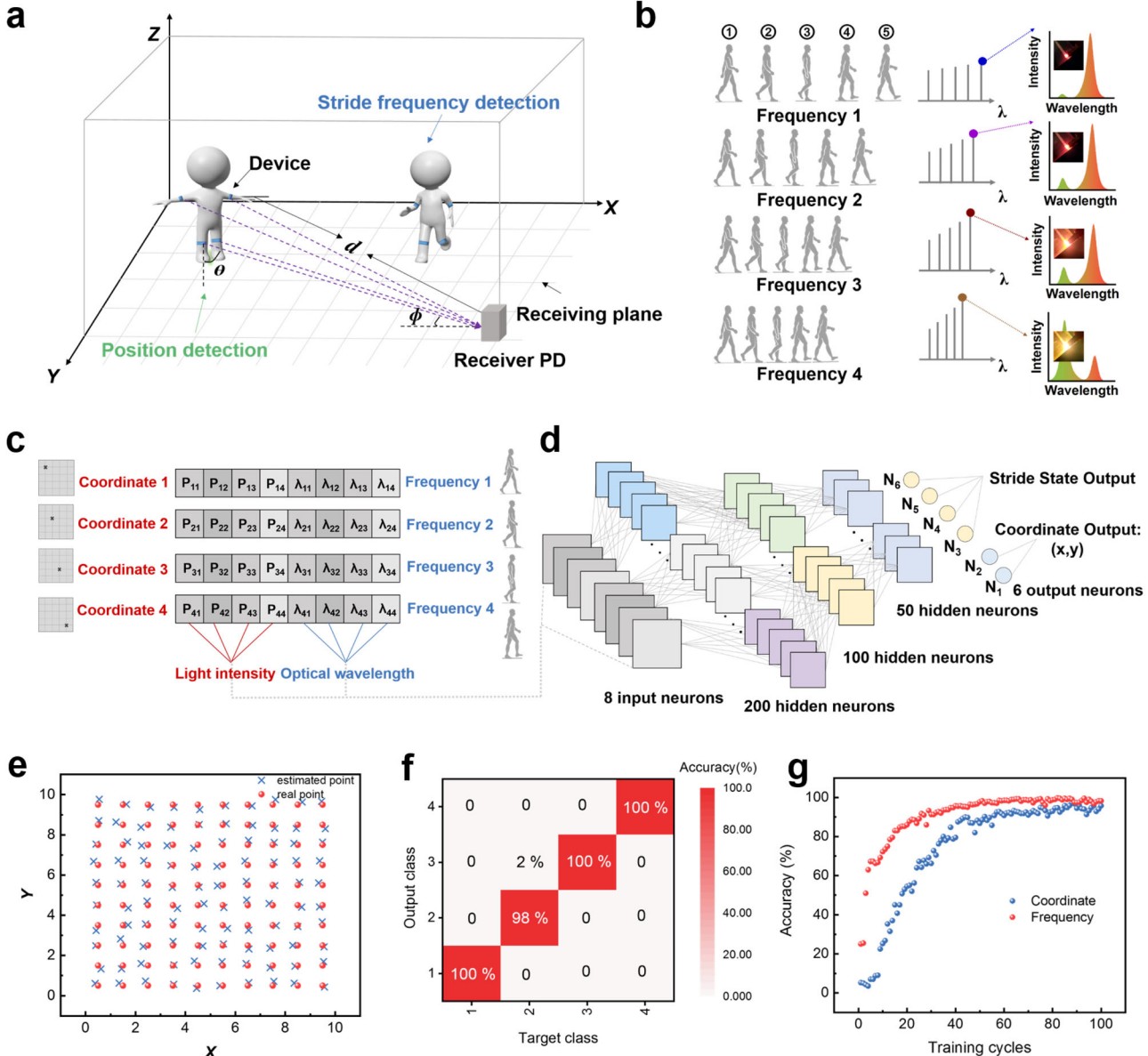

**Fig. 6 | Step frequency-localization multimodal recognition. a** Schematic diagram of SMPS implementation of spatially dynamic step frequency-localization multimodal recognition ($d$ is the distance between the light emission point of the device and the receiver, $\theta$ is the transmit angle relative to the vertical axis of the device, $\varphi$ is the incident angle relative to the receiving axis). **b** Light emission color and wavelength change in four different frequency states. **c** The dataset constructed by the position coordinate and step frequency corresponding to feature vectors $F = [P_{n1}, P_{n2}, P_{n3}, P_{n4}, \lambda_{n1}, \lambda_{n2}, \lambda_{n3}, \lambda_{n4}]$. **d** The process of realizing multimodal recognition by an artificial neural network. **e** Estimated point coordinates and real point coordinates on $10 \times 10$ grid points. **f** The confusion matrix of target label and output label for step frequency recognition. **g** Comprehensive recognition results of frequency-localization recognition.

intensity of the QLED increases with the number of stimuli, indicating the severity of noxious stimuli (Fig. 5g), which can be intuitively reflected by the visible information display.

## Step frequency-localization multimodal recognition

Moreover, a wireless optical communication-based artificial sensory system with TENG and QLED is constructed to realize spatially dynamic step frequency-localization multimodal recognition to further demonstrate the potential of our system in human-computer interaction. As shown in Fig. 6a, flexible TENG will attach to the human limb joints and detect walking signal as a presynaptic spike, then transfer to QLED to generate light output, and the light detector is placed in a suitable position to ensure that it can receive the illumination information emitted by any point in the plane. It is worth noting that the light-emitting synaptic device with memory capacity can generate a response proportional to the frequency change. The color and wavelength of light emitted at the end of the four stride states are affected by the frequency input (errors caused by different exercise intensities are negligible), as shown in Fig. 6b and Supplementary Fig. S19. The plane is divided into $10 \times 10$ grids, the position coordinates of each grid are recorded as label vectors, and the position coordinates can be expressed as $(x_n, y_n)$. By collecting the light information from QLEDs at different grid points, the database feature vector is constructed, and the received signal intensity vector can be expressed as $P = [P_{n1}, P_{n2}, P_{n3}, P_{n4}]$. Depending on the step frequency, light wavelength vectors can be expressed as $\lambda = [\lambda_{n1}, \lambda_{n2}, \lambda_{n3}, \lambda_{n4}]$. When the motion amplitude is fixed, the output light intensity increases as the distance between the human and the detector decreases, and the wavelength of light is determined at the time of emission, so it is not affected by distance (Supplementary Fig. S20). In this way, each position coordinate and

step frequency has a set of corresponding received signal feature vectors $\mathbf{F} = [\mathbf{P_{n1}}, \mathbf{P_{n2}}, \mathbf{P_{n3}}, \mathbf{P_{n4}}, \lambda_{n1}, \lambda_{n2}, \lambda_{n3}, \lambda_{n4}]$, so as to construct the database, as shown in Fig. 6c and Supplementary Fig. S21. We put 400 feature vectors generated from different position coordinates and step frequencies into an artificial neural network with 8 input neurons and perform full connections between neurons. The connections between input neurons and hidden neurons represent the synaptic weights. The output 6 neurons are assigned to the position coordinate output neurons $N_1$, $N_2$ and the stride state output neurons $N_3$, $N_4$, $N_5$, $N_6$, where the final stride state output is obtained by performing the argmax function operation on the output values of neurons $N_3$, $N_4$, $N_5$, $N_6$ (Fig. 6d). Figure 6e depicts the estimated position coordinates on $10 \times 10$ grid point and compares it with the actual position. The criterion of matching degree is defined as $(x - x_0)^2 + (y - y_0)^2 \leq 1$ and the positioning error $r$ is described as $|x - x_0| + |y - y_0|$. The results show that the average positioning error of each point in the test set is 0.2346, and the maximum positioning error $r_{max}$ is 0.4295, which proves that the proposed method is feasible to achieve high-precision positioning. The confusion matrix of the target label and output label in the test set is validated, which shows that the system can realize the recognition of different step frequencies (Fig. 6f). The comprehensive results show that the model has high classification accuracy and robustness after 100 training cycles, and successfully realizes localization and dynamic step frequency recognition, with the accuracy of 98.2% and 99.5%, respectively (Fig. 6g). It is proved that the SMPS with multiparameter information parallel output has great potential in multi-pattern recognition of human–computer interaction.

Finally, we summarize the superiority of the SMPS in the following aspects: (i) Simple in structure and flexible in operation: there are no large optical components or complex modules required to participate in SMPS, and wireless communication overcomes the flexibility limitations of signal transmission and application scenarios. (ii) Strong robustness: SMPS has the capability of wavelength–amplitude multi-parameter information parallel output, avoiding decoding errors caused by the loss of single-dimension information parameters, which can achieve strong fault tolerance in photonic signal transmission. (iii) Visible information display of the multilevel color responses: The SMPS realizes synaptic light output and accompanying visible multiple-color response through sensory stimulation, which shows the multi-level pain warning process of organisms more intuitively than the electrical output or monochromatic light output of previous synaptic devices. (iv) Multi-modal and high accuracy recognition: Without the need for the complex coupled auto-en/decoder and peripheral circuit module in traditional multimodal recognition, the SMPS provides unique tunable multi-wavelength and amplitude outputs, which can simultaneously obtain the human dynamic stepping frequency and spatial positioning information, and achieve efficient multi-modal information recognition with the high accuracy of 99.5% and 98.2%, respectively.

## Discussion

In summary, a sensory memory processing system based on a TENG and QLED is demonstrated, which can realize multi-wavelength synaptic-polychromatic light emission and multimodal information recognition. The optical output signal enables robust information encoding and transmission, intelligent decision processing, and human–machine interaction. Finally, an SMPS-based spatial positioning and dynamic stepping frequency efficient multi-modal recognition were proposed for the first time, which can achieve an accuracy of 98.2% and 99.5%, respectively. Therefore, this work provides a valid strategy for simple components, flexible operation, strong robustness, and a high-efficiency artificial sensory memory processing system, which is crucial for the development of next-generation artificial intelligence interactive equipment and sensory-neuromorphic photonic systems.

## Methods

### Materials
The indium tin oxide (ITO) substrates were obtained from Shenzhen Huanan Xiangcheng Technology Corp. The red and green CdSe/ZnS QDs were obtained from Poly OptoElectronics Co. Ltd. The solution of silver nanowires (AgNWs) (5 mg/mL in isopropanol) was obtained from Suzhou ColdStones Technology Co., Ltd. ZnO nanoparticles (ZnO NPs) were synthesized by a solution method. The cross-linked PVP solution was prepared by mixing 150 mg of Poly(4-vinylphenol) (PVP) powder with 15 mg of 4,4′-(hexafluoroisopropylidene)-diphthalic anhydride in 1 ml of Propylene glycol monomethyl ether acetate solvent.

### Device fabrication
The fabrication process of TENG: PDMS film substrate was prepared by mixing liquid elastomer, and the ratio of base to curing agent was 10:1. The film was annealed at 80 °C for 2 h after spin-coating at 500 rpm for 30 s on the glass. The annealed PDMS thin film was subjected to plasma treatment. Then 5 mg/mL silver nanowires solution was spin-coated on PDMS film at 1000 rpm for 60 s, and the film was peeled off from the glass after annealing at 70 °C for 2 min. In the same way, another annealed PDMS film is stripped from the glass to act as the friction layer. The single-electrode TENG was then fabricated by encapsulating the films together and leading out a copper wire. The fabrication process of QLED: The PEDOT:PSS dispersion solution was spin-coated onto ITO substrate at 3000 rpm 60 s, followed by annealing at 120 °C for 10 min. The Poly-TPD was deposited on PEDOT:PSS film by spin-coating 2000 rpm for 40 s, then annealing at 120 °C for 10 min. The cross-linked PVP solution was spin-coated at 600 rpm for 5 s, then 2000 rpm for 30 s, followed by annealing at 120 °C for 2 h in the glove box. The mixed quantum dot solution was spin-coated at 2000 rpm for 40 s, then annealed at 60 °C for 10 min. Then the ZnO nanoparticle solution was deposited on hybrid QDs film by spin-coating with 3000 rpm for 40 s and subsequently was annealed at 120 °C for 30 min. Finally, the Ag cathode with a thickness of 60 nm was thermally evaporated through a shadow mask.

### Device characterization
The output signals of TENG were measured by Keysight DSOX 1202 oscilloscope. Transient EL measurements of the QLED were detected using a photodetector (EOT Silicon PIN Detector ET-2030) and an oscilloscope (Keysight DSOX 1202). The brightness and EL spectrum were calculated from an Src-200 spectral color luminometer. The electrical characteristics of the synaptic device were measured using Keithley B2902A. The surface morphologies of the film were measured by atomic force microscopy (AFM, Bruker Multimode 8).

## Data availability
The data that support the findings of this study are available from the corresponding author upon reasonable request.

## Code availability
The codes used for the simulations are available from the corresponding author upon request.

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

## Acknowledgements

This work was supported by the National Natural Science Foundation of China (U21A20497 and 61974029) Natural Science Foundation for Distinguished Young Scholars of Fujian Province (2020J06012), and the Fujian Science & Technology Innovation Laboratory for Optoelectronic Information of China (2021ZZ129).

## Author contributions

H.P.C. and T.L.G. conceived the project, L.T.S., Q.Z.C., and R.J.Y. designed and performed the experiments and collected the data. L.T.S., R.J.Y., C.S.G., and L.J.L. analyzed and discussed the data. H.P.C. supervised the project. L.T.S. wrote the paper.

## Competing interests

The authors declare no competing interests.
