## [Peer Review File · Nature Communications]

REVIEWER COMMENTS

Reviewer #1 (Remarks to the Author):

In this article, authors fabricate a sensory memory processing system for multi-modal information recognition. The optical output signal with multi-wavelength synaptic-polychromatic light emission provides a strategy for sensory-neuromorphic photonic systems. This manuscript reports three separate results: 1. the TENG, 2. the QLED, and 3. the application of the combination of these two devices. The third result is dependent on the first two results. It is true that the topical area and content of this manuscript are appropriate, but the authors could perhaps separate out the different topics. One reason for this criticism is that it is hard for this reviewer to understand/critique the neuromorphic computing part of the study. I suspect it will be difficult to find a reviewer conversant (enough) in both TENG, QLED and the details of neuromorphic computing. I restrict comments to the device physics in this manuscript.

1. The SMPS seems to be a simple combination of TENG and QLED. The reviewer cannot see the necessity of this combination. The authors claim that the emitted light has multi-dimensional information, i.e., intensity and wavelength. However, the reviewer believes that the intensity and wavelength have strong correlations because they are both related to injected current (Figure 2,3). So, they are the same information. If the authors believe they are different information, the authors should prove that they are independent with each other.
2. The lack of apparent interest in the QLED performance is highlighted, for example, by the fact that the EQE and efficiencies of the device are not provided. Have you compared the performance of the QLED with the state-of-the-art devices reported before? Can the QLED show comparable or better performance with the state-of-the-art devices? It seems that adding a PVP layer may greatly damage the device performance. The same questions should be asked for the TENG device in this article.
3. I suspect the authors found the neuromorphic computing angle for this system more interesting and seem to have many interesting results. Much time is spent on this part of the paper in a disjointed fashion. Although it is interesting to note that the authors demonstrated two applications using their devices, however, the results are mostly based on simulated experiments, and it is unlikely that these devices can be integrated into an array because of their device structure. The reviewer couldn't find the necessity of your devices in artificial neural network.
4. Stable I-V curves with hysteresis during the measurement cycles are important for artificial synapses, which reflects the stable high- and low-resistance state. But the I-V curves in Figure 3b keep changing throughout the measurement cycles due to the lack of negative voltage region. Please provide the I-V characteristics of the QLED with more measurement cycles to obtain a stable I-V curves (including writing and erasing processes).

5. It is curious that in Figure 3c, the current change is nonlinear with pulse, but in Figure 3d,e, the light intensity change is almost linear. What is the mechanism behind the phenomenon?

In summary, the reviewer is not, unfortunately, an expert in the area of neuromorphic computing so no claim of suitability for publication is made for the neuromorphic results. The device results seem too cursory for publication in Nature Communications. I am sorry but I cannot recommend the publication of this work in Nature Communications.

Reviewer #2 (Remarks to the Author):

The authors report an efficient sensory memory processing system (SMPS), which can process sensory information and generate synapse-like and multi-wavelength light-emitting output. As a result, efficient multi-modal information recognition of dynamic step frequency and spatial positioning have been successfully realized with the accuracy of 99.5% and 98.2%. This proposed system realized diversified utilization of light in information processing and multi-modal information recognition in a simple structure and efficient way, without complex execution module and separation of memory processing based on the traditional CMOS architecture, which has the potential to simplify the circuit units and paves the way for a new generation of intelligent optical communication and neuromorphic sensory system. The paper can be recommended for publication in Nature communications after a minor revision. Here are a few questions.

1. In Figure 2a, the author use two layers of PDMS to fabricate TENG. What is the role of two different PDMS layers? Please provide more detail in the manuscript.
2. As for the characterization of synaptic properties in Figure 3b, the synaptic current curve does not show relaxation properties as in other reported work on artificial synapses. Is this representation correct? Please explain.
3. A quantum-dot light emitting diode (QLED) device is used in the SMPS, what is the difference between QLED and other light-emitting synaptic devices? How about photoelectric synaptic devices to realize utilization of optical information?
4. The authors add a PVP layer to serve as functional layer for hole trapping. Will the PVP block affect the charge transfer? In QLED, the electron transport is better than the hole transport. The authors should provide a comparison of output features with or without the PVP layer.

5. In multi-dimensional information optical communication and visible multi-level injury warning application of Figure 5, the peak in the output that exceeds the given threshold corresponds to '1'..., how is the threshold determined? More detail about this part should be provided.

6. In multimodal recognition application, the authors claim that light-emitting synaptic device with memory capacity can generate a response proportional to the frequency change. Is there any reference or experimental data to support the schematic diagram in Fig. 6b?

Reviewer #3 (Remarks to the Author):

In the paper "A sensory memory processing system with multi-wavelength synaptic-polychromatic light emission for multi-modal information recognition", the authors reported a sensory memory system which can realize multi-wavelength synaptic-polychromatic light emission and multimodal information recognition. Based on this system they demonstrate a multi-level pain warning process. In my opinion, this work is interesting, the proofs are strong enough to support the authors' viewpoints and the experimental procedure is rigorous and accurate. Therefore, the article should be published in Nature Communication after some very minor corrections

Comments:

1. State-of-the-Art: I think you are missing some important reference that are demonstrating for the first time the effect of memristive optical switching and detection. For example:

a. Analog Nanoscale Electro-Optical Synapses for Neuromorphic Computing Applications, Kevin Portner, Manuel Schmuck, Paul Lehmann, Christoph Weilenmann, Christian Haffner, Ping Ma, Juerg Leuthold, Mathieu Luisier, and Alexandros Emboras, ACS Nano 2021 15 (9), 14776-14785, DOI: 10.1021/acsnano.1c04654

b. "Opto-electronic memristors: Prospects and challenges in neuromorphic computing" A Emboras, A Alabastri, P Lehmann, K Portner, C Weilenmann, P Ma et al. Applied Physics Letters 117 (23), 230502

2. When reading the article it was not clear at all if all the components of the suggested system are co-integrated. Could you mention that explicitly in the paper?

3. I would expect to finish your paper with a conclusion. Can you write one?

Response to reviewers:

We thank all the reviewers for their insightful comments on our manuscript. Reviewer's comments are in blue, while our responses are immediately below each comment. Modifications including the non-scientific changes to the manuscript are highlighted in the manuscript itself.

Reviewer #1 (Remarks to the Author):

In this article, authors fabricate a sensory memory processing system for multi-modal information recognition. The optical output signal with multi-wavelength synaptic-polychromatic light emission provides a strategy for sensory-neuromorphic photonic systems. This manuscript reports three separate results: 1. the TENG, 2. the QLED, and 3. the application of the combination of these two devices. The third result is dependent on the first two results. It is true that the topical area and content of this manuscript are appropriate, but the authors could perhaps separate out the different topics. One reason for this criticism is that it is hard for this reviewer to understand/critique the neuromorphic computing part of the study. I suspect it will be difficult to find a reviewer conversant (enough) in both TENG, QLED and the details of neuromorphic computing. I restrict comments to the device physics in this manuscript.

“Author reply”: Thanks for the reviewer's suggestion. More explanation and description have been added in our manuscript, and specific explanations are illustrated below.

As for the SMPS proposed in this paper, which is obtained from the integration of TENG and QLED devices. The characterization of the two devices is independent, while the connection between them is inseparable when they are combined as a system. For the neuromorphic system, it is the demonstration and exploration of the proposed system in the field of artificial intelligence and future human-computer interaction and multi-modal perception.

As the reviewer stated, the third result depends on the first two results. Attribute to the self-powered sensing and wearable, real-time monitoring features of TENG, as well as the utilization of multi-wavelength light emission information by QLED synaptic device, the proposed SMPS can realize the process of capturing sensorimotor information and completing multi-mode recognition concisely and efficiently through the neural network. TENG receives human motion signals and converts them into voltage pulses and supplied to the QLED. The QLED store and memory the received information and then output the postsynaptic response. Due to the unique multi-wavelength light emission characteristics, it can confer the advantage of multiple utilization of optical information. By constructing the database of the neural network using the luminous intensity and wavelength, the collected information is used as the input of the neural network.

Here, our SMPS can be considered as the first stage of data access in the neural network. It can collect, integrate and extract massive sensory data in time, store and

preprocess the collected sensory data, and enable neural networks to have learning and cognitive functions, greatly improving the efficiency of recognition.

Therefore, the system is an essential input terminal and information processing component for neural network, and neuromorphic computing is a practical mean to reflect the system and a new paradigm for the future development of artificial intelligence interactive devices and sensory-neuromorphic photonic system.

Comments:

1. The SMPS seems to be a simple combination of TENG and QLED. The reviewer cannot see the necessity of this combination. The authors claim that the emitted light has multi-dimensional information, i.e., intensity and wavelength. However, the reviewer believes that the intensity and wavelength have strong correlations because they are both related to injected current (Figure 2,3). So, they are the same information. If the authors believe they are different information, the authors should prove that they are independent with each other.

“Author reply”: Thanks for the reviewer’s suggestion. Regarding the question of “The SMPS seems to be a simple combination of TENG and QLED. The reviewer cannot see the necessity of this combination.”

In our SMPS, the self-powered sensor based on TENG realizes the function of sensing human motion information and supplying power to QLED. The light-emitting synaptic device based on QLED could realize neuromorphic operation which is different from the traditional luminous devices. The combination of TENG and QLED could collect and process sensory information and generate synapse-like and multi-wavelength light-emitting output, realizing diversified utilization of light in information processing and multi-modal information recognition. Furthermore, information modulation can be achieved through mechanical motion, and the complexity and energy consumption of traditional circuits can be significantly reduced.

(I) The function of TENG in SMPS: Among SMPS, TENG can transform the external environment information into electrical signals through triboelectrification and electrostatic induction mechanism. The intensity of these signals can be correlated with the intensity of external stimuli, so it can be used as a bridge for the device to interact with the external environment. Due to its high output characteristics and extensive adaptability, it has been developed as an energy harvester to drive wearable sensors. On the other hand, the receptors of most artificial sensing systems are powered by extra voltage capacitors and piezoresistive pressure sensors at present. This requires a huge amount of energy to drive the hundreds of millions of receptors in the neural network and thus creates unnecessary waste. The operation mechanism of TENG has the characteristics of self-power supply. Combining the TENG with neuromorphic devices can significantly solve this energy problem without the need for external power supply access.

In our system, the TENG also takes advantage of the above aspects to realize a multi-purpose, self-powered sensor. In addition, the TENG is prepared into a

single-electrode mode, which is more suitable for biological energy harvesting and wearable use. For example, In Figs. 5f and 5g of the manuscript, artificial nociception receptor is simulated, and TENG can convert external signals into electrical signals as the input of synaptic device through direct stimulation. In Fig. 6 of the manuscript, the SMPS based on TENG can realize the monitoring of human movement and walking status.

(II) The function of QLED in SMPS: A single TENG as a sensor can only detect external signals, without further neuromorphic functions such as storage, integration and memory of these signals. Using neuromorphic device such as synaptic device to build artificial neuromorphic sensory system is to complete the entire process from analog signals input to the human brain for processing. Therefore, the use of the synaptic device in our SMPS is necessary. In addition, compared with the traditional artificial synapses with electrical output, optical signals have the advantages of high-speed transmission, large bandwidth, multiple degrees of freedom, low power consumption and anti-interference. Artificial synapses with optical output can provide more options for multi-channel signal transmission in artificial neural networks. In addition, optical information has rich degree of freedom characteristics, which can be used as carriers to realize information loading. Therefore, it will be a promising solution to apply the multi-information loading characteristic of light to multi-modal recognition field. In our system, the artificial light-emitting synapse prepared by QLED has the characteristic of emitting light with different wavelengths. We use the multi-information loading characteristics of light to apply it in the multi-mode recognition, realizing the dual recognition of spatial position and step frequency.

As a system, SMPS can complete the functions of sensory memory processing including multi-mode recognition, which is not possible without either TENG or QLED. Therefore, for our SMPS, the combination of TENG and QLED is necessary and indispensable.

Regarding the question of “the intensity and wavelength have strong correlations because they are both related to injected current (Figure 2,3). So, they are the same information. If the authors believe they are different information, the authors should prove that they are independent with each other.”

For QLED light-emitting synaptic device, as the reviewer said, its luminescence intensity and wavelength are related to the injected current. When we measure synaptic characteristics, one voltage pulse input corresponds to one luminescence intensity and one wavelength output. Therefore, we agree with the reviewer's comment that the two have a strong correlation. However, when we use the photodetector for the actual measurement, the two are different: **when the input voltage pulse is certain, the received light intensity will change with the distance between the device and the photodetector, while the luminous wavelength of QLED will not change with the distance.**

This is the basis of our Fig. 6 application: The photodetector is placed at a fixed position in space, and SMPS is placed on the surface of the human body. When the human moves at any position in space, the electric signal generated by the TENG is

transmitted to the QLED to generate optical output. When the motion amplitude is fixed, the output light intensity increases with the decrease of the distance between the human and the detector, according to the principle can determine the action of space position coordinates. The wavelength of the light is determined when the light is emitted, so it is not affected by the distance (Supplementary Fig. S20). Therefore, it is regarded as another dimension of information here. By utilizing the frequency dependent characteristic of synaptic device (which can produce a response proportional to the frequency), the results of the light emission color and wavelength generated at the end of the state of different walking step frequency are obtained.

For this process, only QLED device with different wavelengths light-emitting can be completed. If the light-emitting device only contains a fixed wavelength, it will lose this information dimension judged by wavelength. Although light intensity alone can also obtain frequency-dependent output (Fig. 3d, the higher the frequency, the greater the output light intensity), it is affected by the distance from the test detector. Therefore, the light intensity obtained by the actual test is not accurate and cannot be used.

Supplementary Fig. 20. The change of the output light intensity and wavelength with the position. When the motion amplitude is fixed, the output light intensity increases as the distance between human and the detector decreases, and the wavelength of light is determined at the time of emission, so it is not affected by distance.

Therefore, for our SMPS, the separation of light intensity and wavelength must be adopted when collecting the data. By collecting the light information from QLEDs at different points, the database feature vector is constructed, and the received signal intensity vector can be expressed as $P = [P_{n1}, P_{n2}, P_{n3}, P_{n4}]$. Depending on the step frequency, light wavelength vectors can be expressed as $\lambda = [\lambda_{n1}, \lambda_{n2}, \lambda_{n3}, \lambda_{n4}]$. In this way, each position coordinate and step frequency has a set of corresponding received signal feature vectors $F = [P_{n1}, P_{n2}, P_{n3}, P_{n4}, \lambda_{n1}, \lambda_{n2}, \lambda_{n3}, \lambda_{n4}]$, so as to construct the database. Therefore, the authors believe that for the SMPS proposed here, the light

intensity and wavelength are different information, which is attributed to the complete independence of the two in the actual measurements.

Modified manuscript:

Page 12

Multiparameter information optical communication and visible multi-level injury warning

Page 12

Then, the light **multiparameter** information is linearly mapped into sequences. Finally, the sequence is decoded to read the information.

Page 16

It is proved that the SMPS with **multiparameter** information parallel output has great potential in multi-pattern recognition of human-computer interaction.

(ii) Strong robustness: SMPS has the capability of wavelength-amplitude **multiparameter** information parallel output, avoiding decoding errors caused by the loss of single dimension information parameter, which can achieve strong fault tolerance in photonic signal transmission.

Fig. 5. Multiparameter information optical communication and visible multi-level injury warning. (a) Schematic diagram of intelligent optical communication based on wireless transmission and reception. (b) The flow chart for mechanically modulated optical communication. (c-e) The corresponding light amplitude intensity and wavelength (dominant wavelength) readings when encoding/decoding numbers 4, 5, and 9 in optical communications (The shaded parts in the figure represent the loss of (d) amplitude information and the loss of (e) wavelength information). (f) Schematic diagram of SMPS to realize biological pain perception and make multi-level warning process. (g) Under continuous stimulation, the system realizes multi-level injury warning response and visible luminous color change.

Page 15

Depending on the step frequency, light wavelength vectors can be expressed as $\lambda = [\lambda_{n1}, \lambda_{n2}, \lambda_{n3}, \lambda_{n4}]$. When the motion amplitude is fixed, the output light intensity increases as the distance between human and the detector decreases, and the wavelength of light is determined at the time of emission, so it is not affected by distance (Supplementary Fig. S20).

2. The lack of apparent interest in the QLED performance is highlighted, for example, by the fact that the EQE and efficiencies of the device are not provided. Have you compared the performance of the QLED with the state-of-the-art devices reported

before? Can the QLED show comparable or better performance with the state-of-the-art devices? It seems that adding a PVP layer may greatly damage the device performance. The same questions should be asked for the TENG device in this article.

“Author reply”: Thanks for reviewer’s suggestion. We add the performance measurements of QLED device in the supplementary Fig. S6, including brightness, EQE and efficiency.

Supplementary Fig. S6. (a) The voltage-luminance characteristic of the QLED under forward-bias. (b) EQE of the devices as a function of current density. (c) CE and PE of the devices as a function of current density.

We compare the performance parameters of multi-wavelength synaptic-polychromatic light emission QLED proposed in this work with those of multi-color light-emitting QLED devices previously reported (table below). As can be seen from the table, in terms of brightness and power efficiency, the QLED device in SMPS can exceed other work, while the QLED does not show better performance in terms of EQE and current efficiency. Because we added additional charge capture layer PVP in QLED, which cause the decrease of the device performance. However, in order to truly complete the sensory memory processing system, we require a light-emitting device with neuromorphic function, so **we have to add PVP layer to make sure that the device have the ability to update the conductance weight**, which leads to a slight damage to the luminescent performance. **It is worth noting that our light-emitting device is different from traditional QLED, but a light-emitting neuromorphic device with synaptic properties.** Therefore, we focus on the simulation of synaptic plasticity rather than the pursuit of excellent luminescence performance.

Sample	Best luminance [cd m ⁻²]	Best External Quantum Efficiency [%]	Best Current Efficiency [cd A ⁻¹]	Best Power Efficiency [lm W ⁻¹]	References
Multilayer transfer printing R/G QDs	<10000	2.5	/	/	1
High-resolution patterning R/G/B QDs	<10000	5.4	2.3	/	2
Layer-by-layer stacked R/G/B QDs film	<1000	/	4.4	/	3
Tandem-structure R/G/B QLEDs	<10000	2.04	4.75	0.46	4
Mixed R/G/B QDs as light emitting layer	10718	0.65	1.74	/	5
Bilayered Y/B QDs	<10000	0.6	1.4	0.6	6
Full-color-tunable R/G/B QLEDs	<1000	13.3	/	/	7
R/G/B QD-mixed multilayer	23352	10.9	21.8	/	8
Multi-wavelength emission with mixed QDs	59320	3.66	6.27	38.4	This work

The working mechanism of PVP charge trapping layer:

Polymer poly(4-vinyl phenol) (PVP) is a dielectric material, the polar groups contained in the PVP side group contain enormous amount of deep traps that allow charging and discharging carriers upon applied voltage. Therefore, in the previous report, this is the main reason for the hysteresis of OFETs with PVP as a dielectric layer⁹⁻¹¹. Here, we exploit this feature to achieve the characteristics of artificial synapses by embedding capture layer PVP in QLED. Part of the holes can be captured by the PVP layer when bias is applied (I). Then, when the bias is removed, the holes are stored in the PVP layer (II). When bias is applied again, trapped holes are released under the action of an applied electric field (III), thus increasing the conductivity and brightness of the device and achieving a simulation of synaptic plasticity (Supplementary Fig. S16).

Supplementary Fig. S16. The working mechanism of PVP charge trapping layer. Part of the holes can be captured by the PVP layer when bias is applied (I). Then, when the bias is removed, the holes are stored in the PVP layer (II). When bias is applied again, trapped holes are released under the action of an applied electric field (III), thus increasing the conductivity and brightness of the device and achieving a simulation of synaptic plasticity.

In addition, we also compare the performance of TENG (Supplementary Fig. S4). As open-circuit voltage is a typical output characteristic, we list the output open-circuit voltage of the TENG with the same structure and different electrode materials or different friction layer materials¹²⁻¹⁹. The PDMS/AgNWs/PDMS single-electrode TENG have good performance because of the excellent electrical conductivity of silver nanowires and the good electronegativity of PDMS compared with other materials.

Supplementary Fig. S4. Comparison of TENG performance with the same structure, (a) different electrode materials or (b) different friction layer materials. CNT: carbon nanotube; GO: graphene oxide; PVC: Polyvinyl Chloride; PDMS: Polydimethylsiloxane; Kapton: Polyimide; PE: Polyethylene; PET: Polyethylene Terephthalate.

Modified manuscript:

Correspondingly, the well-behaved voltage-luminance curve of the device under forward-bias proves that the device has good luminescence behavior, and the external quantum efficiency (EQE), current efficiency (CE), and power efficiency (PE) of the device are also investigated as a function of current density. (Supplementary Fig. S6).

Page 11

The properties of artificial synapse are achieved by embedding the trapping layer PVP into QLED. In general, PVP is a dielectric material containing deep trap-derived, abundant ionic groups that allow for the trapping and detrapping of charge carriers upon application of a voltage. The working mechanism of PVP charge trapping layer is shown in Supplementary Fig. S16.

Page 7

This result can be attributed to the fact that the increased pressure can achieve more contact area and thus increase the surface charge density, enabling higher voltage output. Moreover, the performance of TENG is compared with TENG with the same structure and different electrode materials or different friction layer materials (Supplementary Fig. S4). The PDMS/AgNWs/PDMS single-electrode TENG exhibits good performance because of the excellent electrical conductivity of silver nanowires and the good electronegativity of PDMS.

3. I suspect the authors found the neuromorphic computing angle for this system more interesting and seem to have many interesting results. Much time is spent on this part of the paper in a disjointed fashion. Although it is interesting to note that the authors demonstrated two applications using their devices, however, the results are mostly based on simulated experiments, and it is unlikely that these devices can be integrated into an array because of their device structure. The reviewer couldn't find the necessity of your devices in artificial neural network.

“Author reply”: Thanks for reviewer's suggestion. For our SMPS, TENG and QLED can be integrated into arrays in terms of device technology, which is not difficult for TENG and QLED with mature manufacturing techniques. It is feasible to integrate them into arrays both from the perspective of the device structure and the existing technology.

Regarding the question of “The reviewer couldn't find the necessity of your devices in artificial neural network.”

(I) The working process and function of SMPS in artificial neural network: For the SMPS proposed in this work, we use the TENG that can be attached to the human body as the motion sensor and the QLED device as the artificial synaptic device. The information generated by human movement is converted into electrical signal through the TENG as presynaptic pulse of QLED and then into optical output signal. The photodetector is placed in a suitable position and fixed. In this process, we tested the

light intensity generated by the body swing at different positions in space (10×10 grid points in total). Since the light intensity measured at different positions is slightly different under the same swing amplitude, we could predict the location where the action occurred. On the other hand, since the luminous wavelength does not change with the measure distance, and the light-emitting synaptic device with memory ability can produce a response proportional to the pulse frequency, we measured the luminous wavelength generated at the end of four step frequency states. Furthermore, we repeatedly measured the luminescence wavelength at each position in four step-frequency states, and constructed a database of light intensity and frequency through the measured data of 100×4 sets (Supplementary Fig. S21). We input the data from the database into a fully connected artificial neural network, which includes one input layer, one output layer and three hidden layers, where the connection between neurons represents synaptic weights. This architecture can be simulated by a memory resistance array (Figure b below). The trained artificial neural network has the function of learning and memory, which can predict and recognize the realization of spatial position and step frequency accurately.

Supplementary Fig. 21. Database of light intensity and frequency through the measured data of 100×4 sets.

Different from the synaptic devices used in neural networks to realize the updating of analog conductivity weights of cross-array hardware architecture, the important functions of SMPS proposed in this work focus on the input of neural networks and execute preprocessing of information (Figure a below). There is also several work reported on this way of working. For example, Yang et al. designed a self-powered artificial retina system by integrating a silicon solar cell with a halide perovskite memristor²⁰. External optical signals are input into the solar array

and converted into electrical signals (presynaptic peaks). Subsequently, the perovskite memristor array receives the presynaptic spikes to generate temporary memory information, realizes the image preprocessing at the hardware level, and inputs the first stage image into ANN to realize the final pattern recognition. Tan et al. reported a distributed multi-sensor and biomimetic synaptic integration structure that can sense, process, and memorize multimodal information, as well as fuse multi-sensor data at the hardware and software levels²¹. It integrates artificial visual, afferent, auditory, olfactory and gustatory sensory neural inputs, thus realizing a biologically inspired multi-sensory neural network.

(II) The importance of SMPS in artificial neural networks: In such device structures, TENG mimics the fundamental properties of natural sensory organs or receptors. At the same time, neuromorphic engineering synaptic devices designed to construct bioinspired cognitive adaptation devices to reproduce the processing/memory capabilities of biological synapses. Integrating these two attributes (sensing and memory) into one device could advance the development of related artificial intelligence. **For the current neural network, they need the input of external information, and our SMPS can be considered as the first stage of data access in the neural network, which can collect, integrate and refine massive sensory data in time, and store, preprocess the collected sensory data. The backend neural network receives the preprocessed data through the dynamic training to modify the connection between neurons, making the neural network has the function of learning and cognition, greatly improving the efficiency of recognition.** Therefore, it is necessary to develop such devices which can capture basic sensorimotor properties and realize sensory memory then apply them in neural networks. Their design, manufacturing, and application research are critical to enabling intelligent and humanized systems that interact directly with humans.

In our SMPS, due to the wearable and interactive characteristics of TENG and the unique luminescence characteristics of QLED, SMPS can be used for non-contact and remote measurement. At the same time, distributed rather than array devices are better at collecting information about different parts of human body, as well as better

collect information about light emission. The multiple utilization of optical information gives its unique advantages, through the use of luminous intensity and wavelength to build a neural network database, concisely and efficiently realize the process of multimode recognition, these operations for traditional devices are difficult to achieve due to their own need to separate and complex sensing module, processing module and processing module (Figure below). **Therefore, as a new type of sensory memory processing system, SMPS presents a unique way of multi-mode recognition and provides a new paradigm for the use of artificial neural networks.**

Modified manuscript:

Page 15

In this way, each position coordinate and step frequency has a set of corresponding received signal feature vectors $F = [P_{n1}, P_{n2}, P_{n3}, P_{n4}, \lambda_{n1}, \lambda_{n2}, \lambda_{n3}, \lambda_{n4}]$, so as to construct the database, as shown in Fig. 6c and Supplementary Fig. S21.

4. Stable I-V curves with hysteresis during the measurement cycles are important for artificial synapses, which reflects the stable high- and low-resistance state. But the I-V curves in Figure 3b keep changing throughout the measurement cycles due to the lack of negative voltage region. Please provide the I-V characteristics of the QLED with more measurement cycles to obtain a stable I-V curves (including writing and erasing processes).

“Author reply”: Thanks for reviewer’s suggestion. We have added the I-V characteristics of the QLED under multiple measurement cycles in the Fig. 3. As can be seen from the figure, during the first few scans, the conductance was in an unstable

rising state, and after 30 cycles, QLED gradually obtained a stable I-V curve due to the addition of the negative voltage region scanning.

Due to the presence of charge trapping layer PVP, our device is different from traditional QLED, endowing it synaptic storage and memory characteristics. Therefore, in the first few scans, the device conductance shows a trend of increasing due to the continuous capture of carriers by defects in PVP. With the increase of the number of scans, the device conductance tends to be saturated. This trend of continuous conductance is consistent with the typical characteristics of analog memristors, indicating that it has good potential in simulating biological synapses.

Modified manuscript:

Page 8

Figure 3b shows the I-V characteristics of the QLED during thirty consecutive positive and negative voltage sweeps (-6→0→6→0→-6 V). As the number of voltage sweeps increases, the conductance of the synaptic device increases continuously and tends to saturation, and finally a stable I-V curve is obtained.

Page 27

Fig. 3. Characterization of synaptic properties of QLED. (a) Schematic diagram of the artificial light-emitting synapse device simulated by QLED. (b) I-V characteristics of the QLED during thirty consecutive voltage double sweeps (-6→0→6→0→-6V). (c) The electrical output synaptic properties of the device at different voltage amplitudes (50 pulse number, 30 ms duration). (d) The response of EPSP to pulse stimulation with different frequencies from 1.21 Hz to 6.84 Hz and (e) numbers from 3 to 11. (f) EPSP and EPSC gain (A_{10}/A_1) (determined by the ratio of the 10th EPSC peak (A_{10}) to the first EPSC peak (A_1)) of the artificial synapse from 1.21 Hz to 6.84 Hz (6 V amplitude, 30 ms duration). (g) The electrical synaptic property simulates the learning and forgetting process in the human brain. (h) The light output characteristic and luminance color change of QLED synapse under continuous stimulation.

5. It is curious that in Figure 3c, the current change is nonlinear with pulse, but in Figure 3d,e, the light intensity change is almost linear. What is the mechanism behind the phenomenon?

“Author reply”: Thanks for the reviewer's suggestion, and we have presented the curve of the actual measured current changing with time at 5 V 30 ms of Figure 3c below (Figure c).

The light intensity has a similar trend to the current output. When measuring the frequency and number dependence (Figure a, b), a relatively small number of pulses are selected (Figure c, State I).

In Figure 3d, e of the manuscript, a relatively small number of pulses are selected (up to 11 pulses), while in Figure 3c of the manuscript, we measured the change of current with the pulse under 50 pulses. Since the light intensity output has a similar trend to the current output, both the light intensity and the current intensity tend to be approximately linear under the action of the first few pulses (Figure a, b). This corresponds to the State I in Figure c. As can be seen from Figure c, the enhancement trend of current can be roughly divided into two stages: in the first stage, the first few pulses, corresponding to the number of pulses given in Figure 3d, e of the manuscript (Figure a, light intensity output) and Supplementary Fig. S11, 12 (Figure b, current output). At this time, the conductance continues to increase in an approximately linear trend as the trap in PVP layer continues to capture the charge. With the increase of time, the trap reaches saturation and no longer captures charge, and the conductance gradually saturates in a non-linear trend, corresponding to State II (as shown in the figure below). This is why the change is nonlinear Figure 3c, while in Figure 3d, e of the manuscript, the change in light intensity is almost linear.

Reviewer #2 (Remarks to the Author):

The authors report an efficient sensory memory processing system (SMPS), which can process sensory information and generate synapse-like and multi-wavelength light-emitting output. As a result, efficient multi-modal information recognition of dynamic step frequency and spatial positioning have been successfully realized with the accuracy of 99.5% and 98.2%. This proposed system realized diversified utilization of light in information processing and multi-modal information recognition in a simple structure and efficient way, without complex execution module and separation of memory processing based on the traditional CMOS architecture, which has the potential to simplify the circuit units and paves the way for a new generation of intelligent optical communication and neuromorphic sensory system. The paper can be recommended for publication in Nature communications after a minor revision. Here are a few questions.

1. In Figure 2a, the author use two layers of PDMS to fabricate TENG. What is the role of two different PDMS layers? Please provide more detail in the manuscript.

“Author reply”: Thanks for reviewer’s suggestion. We have added a detailed explanation of the structure and function of this part to the manuscript. We used two layers of PDMS to make single-electrode TENG, where the effects of the two layers of PDMS are different. We use the PDMS of the upper layer as the friction layer, due to its good electronegativity, can generate charge transfer when separated in contact with the finger, which is called "triboelectrification". Since PDMS have good stretchability and can be used as a flexible substrate to attach to the surface of untitled/human body, we use the PDMS in the lower layer as the substrate of the whole TENG to provide wearable function. This layer does not participate in charge transfer in actual work, and the specific working mechanism is shown in Supplementary Fig. S1. In contact with human skin, due to different electron affinity, the skin and the PDMS friction layer generate equal amount of positive and negative charges respectively. Electrons conduct from skin to PDMS surface due to triboelectric effect (Fig. S1, I). When the skin is separated from PDMS, the charges are unbalanced. In order to neutralize the charges, Ag NWs electrode begins to charge positively, and electrons flow through the ground (Fig. S1, II). When the skin is far enough away from the TENG, the charges reach equilibrium (Fig. S1, III). When the skin is close to PDMS again, electrons flow back to Ag NWs electrode, neutralizing the positive charges of Ag NWs until the skin is full contact with PDMS (Fig. S1, IV). In this way, circulating signals will be generated on the electrode.

Supplementary Fig. S1. Working schematic of the single-electrode TENG. In contact with human skin, due to different electron affinity, the skin and the PDMS friction layer generate equal amount of positive and negative charges respectively. Electrons conduct from skin to PDMS surface due to triboelectric effect (Fig. S1, I). When the skin is separated from PDMS, the charges are unbalanced. In order to neutralize the charges, Ag NWs electrode begins to charge positively, and electrons flow through the ground (Fig. S1, II). When the skin is far enough away from the TENG, the charges reach equilibrium (Fig. S1, III). When the skin is close to PDMS again, electrons flow back to Ag NWs electrode, neutralizing the positive charges of Ag NWs until the skin is full contact with PDMS (Fig. S1, IV). In this way, circulating signals will be generated on the electrode.

Modified manuscript:

Page 6

The effects of the two layers of PDMS are different. The PDMS of the upper layer is used as the friction layer, due to its good electronegativity, which can generate charge transfer when separated in contact with the finger. Since PDMS have good stretchability and can be used as a flexible substrate to attach to the surface of untitled/human body, the PDMS in the lower layer is used as the substrate of the whole TENG to provide wearable function. A continuous voltage response is achieved by contact separation from the skin, which is a process according to contact-electrification and electrostatic induction, and the specific operating principle is shown in Supplementary Fig. S1.

2. As for the characterization of synaptic properties in Figure 3b, the synaptic current curve does not show relaxation properties as in other reported work on artificial synapses. Is this representation correct? Please explain.

“Author reply”: Thanks for reviewer’s suggestion. As the reviewer said, there is no relaxation feature in the description of synaptic properties given in the manuscript, which is a way to facilitate the observation of the changes in synaptic conductance. We present the curve of the postsynaptic current of the device with time after taking the log function in the ordinate (as shown below), when the synaptic attenuation curve can be clearly seen in the figure. Figures 4d and 4e in the manuscript are also obtained in this way, and the decay time τ value is fitted by the decay function. In this way, it is easy to observe the device attenuation and long term memory characteristics. For the synaptic properties in both modes, it is correct, only in different forms. Previous work on unattenuated synaptic current characterization has been reported. For example, Ye et al reported an artificial synapse based on inorganic lead-free AgBiI_4 perovskites, which exhibits typical analog resistive switching (RS) characteristics and synaptic behaviors²². Several functions of biological synapses are realized including long-term potentiation (LTP), long-term depression (LTD), paired-pulse facilitation (PPF) and spike-timing-dependent plasticity (STDP). Yan et al reported a high-performance and low-power consumption memristor based on 2D WS_2 with 2H phase, which can mimic basic biological synaptic functions²³. Pei et al proposed a memristor model with carbon conductive filaments (CFs) based on carbon quantum dots, which can exhibit excellent resistive switching performance²⁴. From the above work, we can see that most of the devices with unattenuated synaptic current characterization are two terminal devices, which also proves that our synaptic characterization is correct.

3. A quantum-dot light emitting diode (QLED) device is used in the SMPS, what is the difference between QLED and other light-emitting synaptic devices? How about photoelectric synaptic devices to realize utilization of optical information?

“Author reply”: Thanks for reviewer’s suggestion. Compared with the previous proposed light-emitting synaptic device, our QLED light-emitting synaptic device are different in the following aspects: First, it has the characteristics of multi-wavelength (multi-color) light emission, while the traditional light-emitting synaptic devices can only emit a single wavelength of light. Therefore, the multiple utilization of optical

information gives it a unique advantage. Through the luminescence intensity and wavelength to build the database of neural network, the multi-mode recognition process is realized simply and efficiently. Second, it has strong robustness. The QLED light-emitting synaptic device has the capability of wavelength-amplitude information parallel output, avoiding decoding errors caused by the loss of single information parameter, which can achieve strong fault tolerance in photonic signal transmission. Third, it has visible information display of the multilevel color responses: The QLED light-emitting synaptic device realizes synaptic light output and accompanying visible multiple-color response through sensory stimulation, which shows the multi-level pain warning process of organisms more intuitively than the electrical output or monochromatic light output of previous synaptic devices.

As to how about photoelectric synaptic devices to realize utilization of optical information, we believe that photoelectric synaptic devices and light-emitting synaptic device are two different devices, and the operation modes and goals are also different. photoelectric synaptic devices simulate synaptic behavior by taking optical information as the input or control terminal of the device. So far, the research on photoelectric synaptic devices mainly focuses on the dual signal input simulation of visual perception process²⁵⁻²⁷. The light-emitting synaptic device is a kind of device with optical information as the output and synaptic plasticity. Synaptic devices with optical output capability can transmit and process data in parallel at the speed of light, providing more options for multi-channel signal transmission in artificial neural networks. Moreover, the introduction of optical signals is expected to reduce the electronic lead density of large-scale integration through on-chip optical networks, thus promoting the development of large-scale artificial neural networks with low crosstalk and high fault tolerance. The light-emitting synaptic device can be used to realize non-contact and remote measurement due to the utilization of light information, and can be applied to related artificial intelligence and human-computer interaction fields, providing a unique paradigm for the use of artificial neural networks.

4. The authors add a PVP layer to serve as functional layer for hole trapping. Will the PVP block affect the charge transfer? In QLED, the electron transport is better than the hole transport. The authors should provide a comparison of output features with or without the PVP layer.

“Author reply”: Thanks for reviewer’s suggestion. We present I-V curves of 0–6 V of QLED with and without PVP layer below, showing that the device without PVP layer has lower turn-on voltage and higher current output, which may be attributed to the fact that the addition of PVP layer hinders the recombination between charges.

However, for a light-emitting synaptic device, the preparation of QLED devices is more focused on the plasticity that can simulate biological synapse rather than the excellent device output performance of QLED. We compare the synaptic performance of QLED with and without PVP layer (Supplementary Figs. S17 and S18). Supplementary Fig. S17 shows that the device conductance increases as a result of the write pulse stimulation. Even after the use of the erase pulse, the conductance of device with PVP layer changes more significantly than that of device without PVP layer. Supplementary Fig. S18 fitted the synaptic decay curves of the two devices, device with PVP layer show significant memory enhancement over device without pvp layer. These results indicate that QLED device with PVP charge trapping layer have better synaptic performance and are more suitable to be integrated into our SMPS as light-emitting synaptic device.

Supplementary Fig. 17. Conductance of a QLED under multiple write-erase pulses. The write pulse stimulation continuously increases the device current, and

after the erase pulse is applied, the device conductance can return to its original state. (a) Conductance change between devices without PVP layer and devices with PVP layer. (b) The waveform of the applied pulse. The write pulse stimulation continuously increases the device conductance and the conductance of the device with PVP layer changes significantly than that of the device without PVP layer even after the application of the erase pulse.

Supplementary Fig. 18. Long-term decay time curves of devices without and with PVP layer (the ordinate is normalized by the log of the current). The shaded area is the memory enhancement of the respective device relative to the initial resting current level.

Modified manuscript:

Page 12

This clearly illustrates that our device has a similar tendency to a long, slow decline in human memory following an initial rapid decline in biological system. **These results indicate that QLED device with PVP charge trapping layer have better synaptic performance and are more suitable to be integrated into our SMPS as light-emitting synaptic device.**

5. In multi-dimensional information optical communication and visible multi-level injury warning application of Figure 5, the peak in the output that exceeds the given threshold corresponds to ‘1’..., how is the threshold determined? More detail about this part should be provided.

“Author reply”: Thanks for reviewer’s suggestion. We are sorry for not being clearly described in the manuscript. We have added further explanation and description in our manuscript. For Figs. 5c-e in the manuscript, we artificially set the threshold value at 0.4 corresponding to the normalized value of the current intensity and the threshold

value at 425 nm corresponding to the wavelength. Since the luminescence device can respond to the mechanical stimulation pulse, the mechanical modulated optical communication process can be performed. The peak value of output exceeding the given threshold corresponds to '1'. Otherwise, it corresponds to '0', which is no information transmission. For Fig. 5g in the manuscript, we simulated an artificial nociceptive system that can realize multi-level injury warning. Here, since three levels of damage can be judged, we divided the two thresholds at 0.4 and 0.7 of the normalized current intensity. When the current intensity is less than 0.4, there is no injury, the device emits red light. When the current intensity is between 0.4 and 0.7, it corresponds to level injury I, orange light. When the current intensity is greater than 0.7, it corresponds to level injury II, green light emitted.

Modified manuscript:

Page 13

Here, we set the threshold value at 0.4 corresponding to the normalized value of the current intensity and the threshold value at 425 nm corresponding to the wavelength. With this concept, the communication sequence can be decoded.

Page 14

In the receptor simulated by TENG, if the input external stimuli are not strong enough or the number is small, the QLED will present low-energy red light emission and low-intensity output, indicating that the external stimuli are harmless (No injury). Here, since three levels of damage can be judged, two thresholds are divided at 0.4 and 0.7 of the normalized current intensity.

6. In multimodal recognition application, the authors claim that light-emitting synaptic device with memory capacity can generate a response proportional to the frequency change. Is there any reference or experimental data to support the schematic diagram in Fig. 6b?

“Author reply”: Thanks for reviewer’s suggestion. We present the postsynaptic current response in the Supplementary Fig. 19. Under different frequencies from a to d, corresponding to the step frequency change from low to high. By observing the amplitude of the end of the last pulse output current, it can be concluded that the higher the frequency, the greater the synaptic response at the end. In other words, light-emitting synaptic device can produce response proportional to the frequency change, which is attributed to the learning and memory characteristics of synaptic device: The higher the frequency, the shorter the interval between pulses, the stronger the memory ability of the device, resulting in a more obvious increase in conductance.

Supplementary Fig. 19. Postsynaptic current in final state at different frequencies. Under different frequencies from a to d, corresponding to the step frequency change from low to high. Light-emitting synaptic device can generate a response proportional to the frequency change.

Modified manuscript:

Page 14

The color and wavelength of light emitted at the end of the four stride states are affected by the frequency input (errors caused by different exercise intensities are negligible), as shown in Fig. 6b and Supplementary Fig. S19.

Reviewer #3 (Remarks to the Author):

In the paper “A sensory memory processing system with multi-wavelength synaptic-polychromatic light emission for multi-modal information recognition”, the authors reported a sensory memory system which can realize multi-wavelength synaptic-polychromatic light emission and multimodal information recognition. Based on this system they demonstrate a multi-level pain warning process. In my opinion, this work is interesting, the proofs are strong enough to support the authors’ viewpoints and the experimental procedure is rigorous and accurate. Therefore, the article should be published in Nature Communication after some very minor corrections

1. State-of-the-Art: I think you are missing some important reference that are demonstrating for the first time the effect of memristive optical switching and detection. For example:

a. Analog Nanoscale Electro-Optical Synapses for Neuromorphic Computing

Applications, Kevin Portner, Manuel Schmuck, Paul Lehmann, Christoph Weilenmann, Christian Haffner, Ping Ma, Juerg Leuthold, Mathieu Luisier, and Alexandros Emboras, *ACS Nano* 2021 15 (9), 14776-14785, DOI: 10.1021/acsnano.1c04654

b. “Opto-electronic memristors: Prospects and challenges in neuromorphic computing” A Emboras, A Alabastri, P Lehmann, K Portner, C Weilenmann, P Ma et al. *Applied Physics Letters* 117 (23), 230502

“Author reply”: Thanks for reviewer’s suggestion. We have added the above works to our manuscript and references.

Modified manuscript:

Page 2

A large number of computational operations such as multiply and accumulate (MAC) in the traditional way have become a serious burden on current central processing units (CPUs)¹. Inspired by the working mode of the human brain, the idea of enabling neural computing at the synaptic level has attracted great interest²⁻¹¹.

Page 4

By combining the optical implementation method with the artificial neural networks (ANN) architecture, is expected to provide a better computing platform for AI. The optical stimulation of memristors is used as an independent signal to trigger a more linear and symmetric switching behavior, which could allow high-density, energy-efficient neuromorphic computing chips.¹⁰

Page 20

10. Portner, K. et al. Analog Nanoscale Electro-Optical Synapses for Neuromorphic Computing Applications. *ACS Nano* **15**, 14776-14785 (2021).

11. Emboras, A. et al. Opto-electronic memristors: Prospects and challenges in neuromorphic computing. *Appl. Phys. Lett.* **117**, 230502 (2020).

2. When reading the article it was not clear at all if all the components of the suggested system are co-integrated. Could you mention that explicitly in the paper?

“Author reply”: Thanks for reviewer’s suggestion. We are sorry for not being clearly described in the manuscript, and all components of the proposed system including TENG and QLED are co-integrated. We revised and provided an interpretation in our manuscript.

Modified manuscript:

Page 10

Figure 4a-c illustrate the working mechanism of the SMPS. TENG is acted as a receptor to collect human motion signals, and QLED is used as a synapse to analyze and process the collected signals. Here, the TENG is integrated with the QLED and the electrode of the TENG is connected to the anode ITO of the synaptic device, delivering the output voltage signal as a presynaptic potential.

3. I would expect to finish your paper with a conclusion. Can you write one?

“Author reply”: Thanks for reviewer’s suggestion. Since the format of Nature Communications requires the conclusion section to be given in the form of “Discussion”, we have summarized the whole paper in the "Discussion" section at the end of the manuscript. The “Discussion” section is the conclusion of this manuscript.

In summary, a sensory memory processing system based on triboelectric nanogenerator (TENG) and quantum-dot light emitting diode (QLED) is demonstrated, which can realize multi-wavelength synaptic-polychromatic light emission and multimodal information recognition. The optical output signal enables robust information encoding and transmission, intelligent decision processing, and human-machine interaction. Finally, a SMPS-based spatial positioning and dynamic stepping frequency efficient multi-modal recognition was proposed for the first time, which can achieve accuracy of 98.2% and 99.5%, respectively. Therefore, this work provides a valid strategy for simple component, flexible operation, strong robustness, and high efficiency artificial sensory memory processing system, which is crucial for the development of next generation artificial intelligence interactive equipment and sensory-neuromorphic photonic systems.

Reference

1. Kim, BH. et al. Multilayer transfer printing for pixelated, multicolor quantum dot light-emitting diodes. *ACS Nano* **10**, 4920-4925 (2016).
2. Kim, GH. et al. High-resolution colloidal quantum dot film photolithography via atomic layer deposition of ZnO. *ACS Appl. Mater. Interfaces* **13**, 43075-43084 (2021).
3. Zhang, JF. et al. Voltage-dependent multicolor electroluminescent device based on halide perovskite and chalcogenide quantum-dots emitters. *Adv. Funct. Mater.* **30**, 1907074 (2020).
4. Zhang, H, Wang ST, Sun XW, Chen SM. All solution-processed white quantum-dot light-emitting diodes with three-unit tandem structure. *J. Soc. Inf. Disp.* **25**, 143-150 (2017).
5. Hames, BC, Mora-Sero I, Sanchez RS. Device performance and light characteristics stability of quantum-dot-based white-light-emitting diodes. *Nano Res.* **11**, 1575-1588 (2018).
6. Kim, JH. et al. Fabrication of a white electroluminescent device based on bilayered yellow and blue quantum dots. *Nanoscale* **7**, 5363-5370 (2015).
7. Mu, G, Rao TY, Chen ML, Tan YM, Hao Q, Tang X. Colloidal quantum-dot light emitting diodes with bias-tunable color. *Photonics Res.* **10**, 1633-1639 (2022).
8. Lee, KH. et al. Highly efficient, color-reproducible full-color electroluminescent devices based on red/green/blue quantum dot-mixed multilayer. *Acs Nano* **9**, 10941-10949 (2015).
9. Lee, CA. et al. Hysteresis mechanism in pentacene thin-film transistors with poly(4-vinyl phenol) gate insulator. *Appl. Phys. Lett.* **89**, 262120 (2006).
10. Chang, CC, Pei ZW, Chan YJ. Artificial electrical dipole in polymer multilayers for nonvolatile thin film transistor memory. *Appl. Phys. Lett.* **93**, 143302 (2008).
11. Xiang, LY, Ying J, Han JH, Zhang LT, Wang W. High reliable and stable organic field-effect transistor nonvolatile memory with a poly(4-vinyl phenol) charge trapping layer based on a pn-heterojunction active layer. *Appl. Phys. Lett.* **108**, 173301 (2016).
12. Zou, Y. et al. A bionic stretchable nanogenerator for underwater sensing and energy harvesting. *Nat. Commun.* **10**, 2695 (2019).
13. Helseth, LE. Interdigitated electrodes based on liquid metal encapsulated in elastomer as capacitive sensors and triboelectric nanogenerators. *Nano Energy* **50**, 266-272 (2018).
14. Yuan, ZQ. et al. Motion recognition by a liquid filled tubular triboelectric nanogenerator. *Nanoscale* **11**, 495-503 (2019).
15. Wu, YH, Luo Y, Qu JK, Daoud WA, Qi T. Liquid single-electrode triboelectric nanogenerator based on graphene oxide dispersion for wearable electronics. *Nano Energy* **64**, 103948 (2019).
16. Zhang, C, Tang W, Han CB, Fan FR, Wang ZL. Theoretical Comparison, Equivalent Transformation, and Conjunction Operations of Electromagnetic Induction Generator and Triboelectric Nanogenerator for Harvesting Mechanical Energy. *Adv. Mater.* **26**, 3580-3591 (2014).

17. Alluri, NR, Raj N, Khandelwal G, Vivekananthan V, Kim SJ. Aloe vera: A tropical desert plant to harness the mechanical energy by triboelectric and piezoelectric approaches. *Nano Energy* **73**, 104767 (2020).
18. Guo, HJ. et al. Self-sterilized flexible single-electrode triboelectric nanogenerator for energy harvesting and dynamic force sensing. *Acs Nano* **11**, 856-864 (2017).
19. Gogurla, N, Roy B, Park JY, Kim S. Skin-contact actuated single-electrode protein triboelectric nanogenerator and strain sensor for biomechanical energy harvesting and motion sensing. *Nano Energy* **62**, 674-681 (2019).
20. Yang, XY. et al. A self-powered artificial retina perception system for image preprocessing based on photovoltaic devices and memristive arrays. *Nano Energy* **78**, 105246 (2020).
21. Tan, HW, Zhou YF, Tao QZ, Rosen J, van Dijken S. Bioinspired multisensory neural network with crossmodal integration and recognition. *Nat. Commun.* **12**, 1120 (2021).
22. Ye, HB, Liu ZY, Han HD, Shi TL, Liao GL. Lead-free AgBiI₄ perovskite artificial synapses for a tactile sensory neuron system with information preprocessing function. *Mater. Adv.* **3**, 7248-7256 (2022).
23. Yan, XB. et al. Vacancy-induced synaptic behavior in 2D WS₂ nanosheet-based memristor for low-power neuromorphic computing. *Small* **15**, 1901423 (2019).
24. Pei, YF, Zhou ZY, Chen AP, Chen JS, Yan XB. A carbon-based memristor design for associative learning activities and neuromorphic computing. *Nanoscale* **12**, 13531-13539 (2020).
25. Seo, S. et al. Artificial optic-neural synapse for colored and color-mixed pattern recognition. *Nat. Commun.* **9**, 5106 (2018).
26. Zhu, QB. et al. A flexible ultrasensitive optoelectronic sensor array for neuromorphic vision systems. *Nat. Commun.* **12**, 1798 (2021).
27. Kumar, M, Lim J, Kim S, Seo H. Environment-Adaptable Photonic-Electronic-Coupled Neuromorphic Angular Visual System. *Acs Nano* **14**, 14108-14117 (2020).

REVIEWERS' COMMENTS

Reviewer #1 (Remarks to the Author):

All of my concerns have been properly addressed and I would like to recommend the acceptance of the revised manuscript in its current version.

Reviewer #2 (Remarks to the Author):

The revised manuscripts nicely addressed all the critiques. It is recommended to publish.

Reviewer #3 (Remarks to the Author):

My concerns have been addressed in the revisions.